# A Non-Linear Ranking Surrogate based Stochastic Bandits for top-m arm Selection

## Abstract

The top-m arm selection problem has multiple applications, particularly in example selection for enhancing in-context learning in Large Language Models (LLMs). Existing approaches assume a linear relationship between features and rewards, which limits their ability to capture the complex reward landscapes induced by LLMs. Moreover, they typically perform static task-level selection, choosing subsets once offline, which can fail to generalize to unseen queries. This motivates the need for learning a surrogate that can be employed, for instance-level ranking of exemplar subsets. To address these challenges, we formulate the top-m arm selection as a learning-to-rank problem and propose GRASS (**G**ap-indexed bandits with **RA**nking-based non-linear **S**urrogate for **S**election). It is a novel gap-index bandit framework with *non-linear differential sorting* based surrogate to model the scores of the example subsets (arms) for the top-m arm (example subset) selection problem. The nonlinear surrogate is learned offline using gap-index framework with challenger arm sampling to clearly distinguish borderline arms in a fixed-confidence setting and also provides top-m examples. Hence, it can be used in a task-level or instance-level setting. GRASS is as sample-efficient as linear bandit variants, while providing performance gains of **9.4-15.2**% in smaller open-source LLMs while converging faster (**2.35** $x$) than existing state-of-the-art approaches.

## 1 Introduction

Selecting representative instances from a large pool is a recurring need across applications such as drug repurposing, domain adaptation, and few-shot learning. One principled way to study this is as a *subset selection* problem: choosing a small set of examples that captures the task's salient structure according to a task-specific reward signal. A dominant class of methods formulates subset selection as *top-m arm identification* in multi-armed bandits (MABs), where each candidate subset is an arm and rewards come from the task itself Réda et al. (2021). The *top-m arm identification* formulation provides a natural and statistically grounded way to address subset selection: each $k$-sized subset corresponds to an arm, and the goal is to identify the $m$ best arms that maximize validation performance.

In this paper, we focus on *in-context learning* (ICL) with large language models (LLMs) as a concrete and representative instance of subset selection. LLMs can solve new tasks when given a small sequence of demonstrations $(v, w)$ or $(v, \text{rationale}, w)$ in context Brown et al. (2020). However, naively choosing demonstrations (randomly or heuristically) performs poorly Purohit et al. (2024); Li & Qiu (2023). Moreover, ICL performance depends on *subset interactions*—the joint effect of examples shown together—making the problem inherently combinatorial. Casting ICL demonstration choice as top-$m$ arm identification provides a principled route to select representative subsets using the task's own reward, while keeping evaluation budget in check.

Principled example selection approaches can be categorized as either *task-level (static)* or *instance-level (dynamic)*. Static selection chooses a representative set of examples once per task and reuses it at inference time Purohit et al. (2024; 2025a); Li & Qiu (2023). This approach is efficient, but can fail when new queries require reasoning skills absent from the fixed set. Dynamic selection chooses examples per test query, which improves flexibility but is computationally expensive, since evaluating candidate subsets involves searching through an exponentially large space of combinations. Recent work, such as CASE Purohit et al. (2025a), builds upon gap-index bandit frameworks like GIFA Réda

et al. (2021), introducing challenger sampling mechanisms to handle the large search space. While effective, these approaches have two major limitations. First, they rely on a *linear surrogate* $\mathcal{F}_\theta$ to map arm features to rewards. Linear surrogates cannot capture the complex, non-linear dependencies between subsets and task performance in ICL. Importantly, simply substituting a non-linear surrogate is insufficient: the *gap-index framework itself must be adapted* so that index computations remain valid in the non-linear setting. Second, CASE is primarily designed for static selection and does not generalize well to dynamic settings, where runtime selection is required.

**Our contributions.** To overcome these limitations, we propose GRASS (**G**ap-indexed bandits with **RA**nking-based non-linear **S**urrogate for **S**election), a new bandit framework for top-$m$ subset selection. Unlike prior work, GRASS makes the following advances:

- **Non-linear surrogate within gap-index bandits.** We extend the gap-index framework itself to incorporate a non-linear surrogate, ensuring that gap-index computations remain valid beyond linear parametrizations. We further provide **theoretical guarantees**, including bounds on pairwise gap error and on sample complexity.

- **Unified support for static and dynamic selection.** Apart from identifying top-$m$ example subsets at convergence, the surrogate in GRASS is trained offline with gap-index arm comparisons, enabling it to distinguish between borderline arms in a fixed-confidence setting. Once trained, it can be used efficiently at inference time to rank example subsets, supporting both static (task-level) selection and dynamic (instance-level) selection for unseen queries.

- **Efficiency with sample complexity guarantees.** Our approach retains the sample efficiency of linear gap-index bandit variants, while substantially improving performance (**9.4–15.2%**) on smaller open-source LLMs, while preserving sample efficiency. By replacing expensive human annotations or ad-hoc relevance signals with LLM feedback during training, GRASS provides an efficient mechanism to learn a ranking-based surrogate for subset selection.

## 2 RELATED WORK

**Top-m arm identification in stochastic bandits and linearity assumption.** The objective of top-$m$ arm identification is to identify those arms with highest means preferably in a sample efficient manner. While fixed-confidence (Kalyanakrishnan et al., 2012) and fixed-budget settings (Bubeck et al., 2013) exist, our focus is the fixed-confidence setting, where the error probability to estimate the top-$m$ arms should be smaller than a predefined parameter $\delta \in (0, 1)$. Adaptive sampling algorithms like UGapE (Gabillon et al., 2012) and LUCB (Kalyanakrishnan et al., 2012), along with uniform sampling methods (Kaufmann & Kalyanakrishnan, 2013; Chen et al., 2017), have been introduced for the fixed confidence setup, but they lack efficiency in terms of sample complexity. While efficient adaptive sampling methods for linear bandits, such as Fiez et al. (2019), RAGE Zhang et al. (2023), LTS Jedra & Proutiere (2020), PEPS Li et al. (2023), LinGapE (Xu et al., 2017) and LinGame Degenne et al. (2020), have been proposed, they primarily address best-arm identification ($m = 1$). GIFA (Réda et al., 2021) was the first unified framework for efficient top-$m$ arm identification, but requires significant number of gap-index computations and comparisons, leading to high sample complexity. CASE Purohit et al. (2025a) proposes to solve this by proposing principled sampling and creation of challenger shortlists but still assumes a linear relationship between arm features and rewards. However, this does not reflect practical scenarios where the relationship is non-linear. Our proposed work aims to bridge this gap by casting top-m arm selection as a ranking task and employs a non-linear ranking based surrogate to model the arm feature to reward relationship.

**Exemplar Selection for ICL.** The rise of LLMs has transformed them into general-purpose answering engines through emergent capabilities like ICL (Brown et al., 2020; Wei et al., 2022; 2023; Wang et al., 2023; Kojima et al., 2023; Chen et al., 2022) where a few examples are provided to LLMs to demonstrate the task. To eliminate manual selection, several automated methods have emerged, such as reinforcement learning (Zhang et al., 2022; Lu et al., 2023), trained retrievers Xiong et al. (2024), Determinantal Point Processes (Ye et al., 2023a) and constrained optimization (Tonglet et al., 2023). Additionally, instance-level selection methods that are learning-free, such as similarity-based (Rubin et al., 2022), complexity-based (Fu et al., 2023), and MMR (Ye et al., 2023b), have been explored. However, instance-level methods increase inference-time computational costs. To address this, a pre-selected, representative set of exemplars is chosen for ICL, akin to coreset selection methods (Guo et al., 2022), though the key difference is that ICL does not involve parameter updates. While CASE

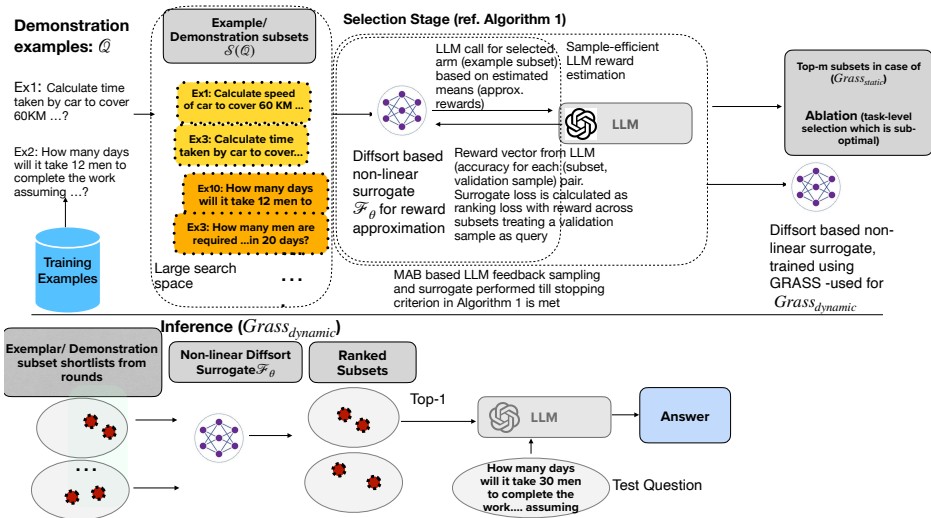

Figure 1: An overview of the proposed example subset selection algorithm which also fits a non-linear surrogate adaptively to clearly separate top-scoring subsets from sub-optimal arms through the proposed gap-index based top-m identification algorithm - GRASS. Please note that rewards are sampled at instance-level (each validation sample) which contributes to reward vector that models scores of subsets for each validation sample. Each arm pull contributes to a subsets with corresponding reward with respect to each validation sample contributing to training samples for non-linear surrogate. The surrogate is updated by ranking loss as indicated in Lines 311-313

Purohit et al. (2025a) and EXPLORA Purohit et al. (2024) aims to proposes bandit-based task-level example selection algorithm they assume the arm feature to reward relationship is linear which is not always practical with respect to LLM rewards. Our proposed approach focuses on non-linear surrogate to model LLM based rewards and offers a *general framework* to integrate task-level and instance-level/dynamic example selection.

**Learning to Rank and Differentiable sorting approaches.** The existing works from online learning to rank literature are somewhat related to the choice of our surrogate Zoghi et al. (2017); Grotov & De Rijke (2016); Li et al. (2019), which learn the parameters of ranking models from user interaction or click data. However, our approach differs fundamentally from this line of work. Unlike these methods, we do not rely on direct user feedback or address challenges like prioritizing or de-biasing rank-sensitive clicks. Moreover, our framework employs an efficient version of gap-index class of algorithms to arrive at top-m arms coupled with efficient learning of a differentiable sorting surrogate that can be employed for online ranking of sets. This allows it to scale efficiently to large search space compared to learning-to-rank models.

## 3 METHODOLOGY

### 3.1 PROBLEM DEFINITION

In ICL, the model processes a sequence of input–output demonstrations followed by a new test input, and is expected to generate the corresponding output. We introduce the problem of ICL as a *subset-selection problem* of choosing representative examples from a large dataset of existing examples Purohit et al. (2024), demonstrations, or training instances Ho et al. (2020). We further show that this subset-selection problem can be modeled formally as a *top-m arm identification problem* in the multi-armed bandit framework. Note that our proposed framework is *generic* and applies broadly to any subset selection problem – where one must select subsets from a large collection of candidates. For concreteness, in this paper we focus on *in-context learning* (ICL) with large language models (LLMs), where the construction of effective contexts is a central challenge. Figure 1 shows an overview of the problem and proposed framework. Our framework primarily focuses on sample-efficient fitting of non-linear surrogate which is employed for instance-level (dynamic)

---

**Algorithm 1** GRASS: Challenger-Aware Surrogate Ranking

---

**Input:** $Q$ (training exemplars); $k$ (prompt size); $S$ (all $k$-subsets of $Q$); $m$ (TOP-m target); $m'$ (challenger size);
    $G$ (data generator); $N$ (query size); $\mathcal{F}_\theta$ (LTR model)
**Output:** $\mathcal{F}_\theta$ (trained surrogate) and estimated TOP-m set

---

1:   INITIALIZE
2:     $U_0 \leftarrow$ random $m$ arms from $S$                                         ▷ Current TOP-m
3:     $C_0 \leftarrow$ the next best $m'$ arms where $m' < m$ (is resampled every iteration)      ▷ Challengers
4:     $\mathcal{D} \leftarrow 0$ or random $M$ subsets from $G(Q)$ to solve cold-start problem
5:     $t \leftarrow 1$

6: **while** $\neg (B_t(ch_t, b_t) \leq \epsilon)$ **do**

7:     (A) IDENTIFY ARMS TO SWAP
8:       $m_t \leftarrow \arg\min_{a \in U_{t-1}} \hat{\rho}_t(a)$                                                    ▷ Weakest in TOP-m
9:       $c_t \leftarrow \arg\max_{a \in C_{t-1}} \hat{\rho}_t(a)$                                            ▷ Strongest challenger
10:    (B) BORDER UPDATE (SWAP IF NEEDED)
11:    **if** $\hat{\rho}_t(m_t) \geq \hat{\rho}_t(c_t)$ **then**
12:         Exchange $m_t$ and $c_t$ between $U_{t-1}$ and $C_{t-1}$
13:    **end if**
14:       $U_t, C_t \leftarrow$ updated sets
15:    (C) EXPAND CANDIDATE POOL
16:       $M_t \leftarrow$ random $m'$ arms from $(U_t \cup C_{t-1})^c$
17:       $C_t \leftarrow \text{top}_{m'}(M_t \cup C_{t-1}; \hat{\rho}_t)$
18:    (D) RECOMPUTE AMBIGUITY FRONTIER
19:       $b_{t+1} \leftarrow \arg\max_{b \in U_t} \max_{ch \in C_t} B_t(ch, b)$
20:       $ch_{t+1} \leftarrow \arg\max_{s \in C_t} B_t(ch, b_{t+1})$
21:    (E) ACQUIRE FEEDBACK & RETRAIN
22:       $a_{t+1} \leftarrow$ selection_rule$(U_t, C_t)$
23:       $r_{t+1} \leftarrow R(\psi(a_{t+1}), \mathcal{V})$                                        ▷ LLM/environment call
24:       $\mathcal{D} \leftarrow \mathcal{D} \cup G(r_{t+1}, a_{t+1})$
25:       Retrain $\mathcal{F}_\theta$ for one epoch on $\mathcal{D}$
26:       $t \leftarrow t + 1$
27: **end while**
28: **Return** $\mathcal{F}_{\theta^*}$ and $U_t$

---

selection. In summary, **Section 3.1.1** provides preliminaries of the example subset selection which is mapped to top-m arm selection in **3.1.2** while discussing limitations of existing static selection or static-dynamic (hybrid) approaches that adopt this formulation. Section **3.2** proposes a novel bandit algorithm to overcome these limitations also aids in learning a non-linear surrogate that can rank subsets at inference time (dynamic). The bandit algorithm we propose is based on gap-index based philosophy Réda et al. (2021) due to it's sample-efficiency and sufficient sampling of ambiguous arms. But ours is a completely different algorithm that accounts for instance-level selection and also accounts for violation of linear assumptions made by existing approaches employing classical linear stochastic bandit algorithms.

### 3.1.1   SUBSET SELECTION FOR IN-CONTEXT LEARNING

In ICL, a large language model processes a sequence of *demonstrations* (i.e., input–output pairs) followed by a test input, and is expected to generate the corresponding test output. We denote each input by $v$ (e.g., a natural language question, a math problem, or a sentence to translate) and each output by $w$ (e.g., an answer, solution, or translation). In some tasks, $v$ may also contain additional reasoning such as rationales or chain-of-thought annotations; for example, solving math word problems may require showing intermediate steps, while translation tasks typically do not.

Let $Q = \{(v_i, w_i)\}_{i=1}^n$ be a pool of $n$ candidate demonstrations, and let $(v_{test}, w_{test})$ denote a test instance. To perform inference on $v_{test}$, we select a subset $S \subseteq Q$ of $k$ demonstrations and concatenate them with the test input to form the context

$$E = \big[(v_{i_1}, w_{i_1}), \ldots, (v_{i_k}, w_{i_k}), v_{test}\big], \quad \hat{w}_{test} = \mathbb{P}_{LLM}(\cdot \mid E).$$

The quality of the chosen subset $S$ has a direct impact on LLM performance. However, a single $k$-subset is rarely sufficient: different subsets capture different reasoning skills or topical knowledge required for a query at test-time, and multiple subsets may yield comparable outcomes (on average across test queries for task-level selection). This motivates the identification of the *top-m subsets* of demonstrations that are most useful for ICL. Formally, let $\mathcal{S}(Q)$ denote the set of all $k$-sized subsets of $Q$, and let $\mathcal{V} = \{(v_{val}(j), w_{val}(j))\}_{j=1}^{n'}$ be a validation set. The objective is to identify $\{a_1, \ldots, a_m\} \subseteq \mathcal{S}(Q)$ that maximize expected performance on $\mathcal{V}$.

The inference procedure with top-$m$ subsets can then be written as:

$$E = \phi(\{a_1, \ldots, a_m\}, v_{test}), \quad \hat{w}_{test} = \mathbb{P}_{LLM}(\cdot \mid E), \tag{1}$$

where $\phi$ constructs the context from selected subsets. For task-level selection, $\phi$ may pick the subset with the lowest validation loss on $\mathcal{V}$. Alternatively, $\phi$ can act as a similarity-based retriever, choosing the most relevant subset from the offline top-$m$ list at inference time, resulting in a hybrid static–dynamic selection scheme Purohit et al. (2024).

Nevertheless, static subsets selected offline may fail to generalize to unseen queries, while fully online selection is computationally prohibitive since running bandit algorithms to convergence per test instance incurs high latency. This motivates the need for a principled formulation that is both *expressive*, capturing complex dependencies in LLM rewards, and *efficient*, supporting practical inference-time selection.

### 3.1.2 TOP-$m$ ARM SELECTION

We formalize the subset selection problem as a *top-m arm identification task* in a multi-armed bandit (MAB) setting. Each candidate subset $a_i \in \mathcal{S}(Q)$ is treated as an arm, with reward defined by LLM performance on $\mathcal{V}$. Since $|\mathcal{S}(Q)|$ can be large, the search space $\mathcal{S}(Q)^m$ of all possible $m$-subsets is extremely challenging.

Let the true reward for an arm $a$ be $\rho(a; \theta^*) = \mathcal{F}_{\theta^*}(x_a)$, where $x_a \in \mathbb{R}^n$ are arm features and $\theta^*$ are the true parameters. Each evaluation of an arm yields a noisy observation:

$$\hat{\rho}(a; \theta) = \rho(a; \theta) + \eta, \quad \mathbb{E}[e^{\lambda \eta}] \leq \exp\left(\frac{\lambda^2 \chi^2}{2}\right),$$

where $\eta$ is sub-Gaussian with variance $\chi^2$.

The top-$m$ identification objective is to output $\widehat{S}_m$ such that

$$\mathbb{P}\left(\widehat{S}_m \neq S_m^\star\right) \leq \delta, \quad S_m^\star = \{1, 2, \ldots, m\},$$

while minimizing the number of samples.

**Gap-index methods and their limitations.** Gap-index bandit algorithms such as GIFA Réda et al. (2021) address top-$m$ arm identification by iteratively estimating arm parameters and comparing the most *ambiguous arms* using gap indices. CASE Purohit et al. (2025a) improves efficiency by sampling a smaller challenger shortlist instead of performing all pairwise comparisons. However, both GIFA and CASE rely on a *linear surrogate* $\mathcal{F}_\theta$, which is restrictive when modeling rewards induced by LLMs. Linear surrogates cannot capture complex, non-linear dependencies between subsets and task performance. Importantly, simply substituting a non-linear surrogate is insufficient: the *gap-index framework itself must be adapted* so that index computations remain valid in the non-linear setting. In GRASS, we provide this modification, extending gap-index bandits with a non-linear differentiable sorting surrogate with theoretical bounds on pairwise gap error and sample complexity. This enables expressive modeling of LLM subset rewards while retaining the statistical efficiency of gap-index methods.

### 3.2 GRASS: A GAP-INDEX ALGORITHM WITH NON-LINEAR RANKING SURROGATE FOR TOP-M ARM SELECTION

Based on above discussion of the problem setup we propose a gap-index based bandit algorithm GRASS with a non-linear ranking surrogate. Since, the task can also be viewed as learning to rank

(LTR) the example subsets (arms), we adopt a *differentiable sorting* model Swezey et al. (2021) as the non-linear surrogate. This surrogate learns to approximate the rewards of arms during the offline run of the bandit algorithm and hence can be used during runtime to rank all example subsets for dynamic selection as shown in Figure 1.

At each step, an exemplar subset (arm) is regarded as a document whose current empirical mean is estimated by $\mathcal{F}_\theta$. $\mathcal{F}_\theta$ is a multi-layer connected network architecture with RELU activations.

$$\hat{\rho}(a_i) = \mathcal{F}_\theta(x_a) = \frac{1}{n'} \sum_{i=1}^{n'} (\mathcal{F}_\theta(x_a, v_{val}(i))), \quad x_a = \left[ \mathcal{H}(v_{val}(i)), \frac{1}{k} \sum_{l=1}^{k} \mathcal{H}(v_{i_l}; w_{i_l}) \right] \quad (2)$$

where $v_{val}(i)$ from $\mathcal{V}$ is treated as a query and the example subset (arm) is treated as the document. We average the sentence embeddings of examples obtained using an encoder $\mathcal{H}$ in the arm to provide a single feature representation for the arm $x_a$ of dimension $d$. The representation combined with query embedding is used as input to the ranking based surrogate $\mathcal{F}_\theta$ to obtain empirical mean estimate.

Then the arm (example subset) being played provides the score based on LLM output on multiple validation samples from $\mathcal{V}$ as rewards.

$$\mathcal{R}(\psi(a_i), v_{val}(i)) = \gamma \left( \mathbb{P}_{LLM}(\cdot \mid \psi(a_i), v_{val}(i)) \right) \quad (3)$$

Here $\gamma$ could indicate accuracy or other relevance measures like BertScore which compares the generated output from LLM $\hat{w}_{val}$ with ground truth $w_{val}$ and outputs a relevance score. And $\psi$ denotes the context / prompt generator function based on given subset of examples and the query to be answered. Based on above definition of LLM performance, $\phi = \frac{1}{n'} \sum_{j=1}^{n'} \mathcal{R}(\psi(a_i), v_{val}(j))$ $\phi$ in Equation 1 could choose a subset of examples that lead to lowest validation accuracy from top-$m$ subsets. Hence reward for an arm can be obtained as: $r(a_i) = \mathcal{R}(\psi(a_i), \mathcal{V}) = [\mathcal{R}(\psi(a_i), v_{val}(i))..\mathcal{R}(\psi(a_i), v_{val}(n'))]$ which is then employed to update the surrogate as detailed below.

An overview of the gap-index based bandit algorithm with differentiable sorting surrogate is shown in Algorithm 1. First a shortlist of good arms $U_0$ is initialized to random $m$ arms in Steps 1-3. The dataset $\mathcal{D}$ which is used to fit the non-linear ranking surrogate is initialized to empty set or random data to solve the cold start problem (Line 4). The the updated $U_t$ is computed by selecting the worst-arm in $U_t$ with lowest empirical mean in current step and swap it with the best challenger arm $ch_t$ in the challenger shortlist $C_t$ (Lines 8-14). The empirical means for above steps are computed using the formulation in Equation 2. In Lines 15-17, we uniformly sample $m'$ arms from $(U_t \cup C_{t-1})^c$, to generate the set $M_t$, and then select the top-$m'$ arms from $M_t \cup C_{t-1}$ to generate the updated $C_t$. The most ambiguous arms $b_t$ (guess for m-best arm) and $ch_t$ (a potentially misassessed arm m-best arm) which determine the stopping criterion are computed with help of gap-indices as shown in Steps 18-20. The gap-index between any two arms $i, j$ is computed as: $B_t(i, j) = \hat{\rho}_t(i) - \hat{\rho}_t(j) + W_t(i, j)$. Here in gap-index computation, $W_t(i, j)$ is computed as per **Equation 6** ( the RHS of the inequality from **Theorem 1**) accounting for the non-linear ranking based surrogate in GRASS. $C_t \cup U_t$ bounds the amount of comparisons required for gap-index computations unlike GIFA. The intuition here is that once the gap-index between most ambiguous arms approaches $\epsilon$ (Line 6), there is no confusion between the empirically estimated top-$m$ arms and it's closest competitor in the challenger shortlist. Then we employ a greedy selection rule Réda et al. (2021), where the arm that minimizes the variance between $b_t$ and $ch_t$ is selected. The error in the empirical mean estimates with respect to rewards is computed by the version of loss $\mathcal{L}(\vec{\hat{\rho}}, \vec{r}) = -\widehat{NDCG}(\vec{\hat{\rho}}, \vec{r})$ employing a relaxed version of NDCG metric Swezey et al. (2021). Here $\vec{r}$ and $\vec{\hat{\rho}}$ are rewards and empirical mean vectors (across arms) for a single query (validation sample). Then through a Stochastic Gradient Descent (SGD) step the surrogate is updated to better estimate the empirical mean ((Lines 22-25)). Hence, apart from top-$m$ subsets (arms) selection, our approach also provides an **efficient mechanism** to learn a differentiable sorting model based on LLM feedback than unlike traditional LTR settings.

### 3.3 SAMPLE COMPLEXITY BOUNDS FOR GRASS

Following Réda et al. (2021), we obtain a high probability $(1 - \delta)$ upper bound on sample complexity of GRASS which is non-trivial and different from linear MAB variants. To derive the same, we first need to define a condition / event on gap indices.

**Definition 1.** *(Good Gap indices)* $\mathcal{E} \triangleq \bigcap_{t>0} \bigcap_{i,j \in [K]} \left( \rho_i - \rho_j \in [-B_t(j,i), B_t(i,j)] \right),$

with $\mathbb{P}(\mathcal{E}) \geq 1 - \delta$ which denotes that a good choice of gap indices $B_t(i,j)$ satisfies event $\mathcal{E}$ with probability greater than or equal to $1 - \delta$. For the above event to hold, it is essential to prove the following bound on pairwise-gap error

**Theorem 1.** *In a fixed-confidence setting, $\delta \in (0,1)$, with probability at least $1 - \delta$, for all pairs $i, j \in \mathcal{A}$:*

$$\left| (\hat{\rho}_t(i) - \hat{\rho}_t(j)) - (\rho(i) - \rho(j)) \right| \leq c_t \sqrt{\frac{2\widehat{V}_{ij,t} \log(2K^2/\delta)}{N}} + \varepsilon_t^{stab} + b_i + b_j + \frac{2M \log(2K^2/\delta)}{3N}. \quad (4)$$

*where $\widehat{V}_{ij,t}$ is the empirical variance of MC-dropout differences, that is for N stochastic predictions (MC dropout forward passes) using the validation set $\mathcal{V}$ $\bar{d}_{ij} := \frac{1}{N} \sum_{k=1}^{N} \left( y_i^{(k)} - y_j^{(k)} \right), \widehat{V}_{ij,t} := \frac{1}{N} \sum_{k=1}^{N} \left( y_i^{(k)} - y_j^{(k)} - \bar{d}_{ij} \right)^2 ., \varepsilon_t^{stab}$ is the SGD stability error, and $b_i, b_j$ are surrogate approximation biases.*

**Theorem 1 Proof structure**: Further details are presented in Appendix B.

Given event $\mathcal{E}$ holds, we derive the sample complexity as follows,

**Theorem 2.** *For GRASS, on event $\mathcal{E}$ on which the algorithm is $(\varepsilon, m, \delta)$-PAC, stopping time $\tau_\delta$ satisfies $\tau_\delta \leq \inf\{u \in \rho^{*+} : u > 1 + H^\varepsilon(\rho) \frac{log(2K/\delta)}{N} + O(K)\}$, where, for algorithm with the largest variance selection rule [1] : $H^\varepsilon(\rho) \triangleq 18c_t^2 \sum_{a \in [K]} \sigma_{a,t}^2 \cdot \max\left\{ \varepsilon^{-2}, \left( \frac{\varepsilon + \Delta(a)}{3} \right)^{-2} \right\},.$*

**Theorem 2 Proof**: On event $\mathcal{E}$, we first demonstrate that the Lemma 1 below holds. Then using stopping criterion and Lemma 1 we derive the upper bound on sample complexity. The detailed proof is available in Appendix E.2. The bound holds for arms in $U_T \cup C_T$. It implies that the top-m arms from $U_T \cup C_T$ are present in $U_T$ with prob. $1 - \delta$, if $T > \tau_\delta$, and $K$ is the size of $U_T \cup C_T$.

**Lemma 1.** *On the event $\mathcal{E}$, for all $t > 0$,*

$$B_t(ch_t, b_t)(t) \leq \min(-(\Delta(b_t) \vee \Delta(ch_t)) + 2W_t(b_t, ch_t), 0) + W_t(b_t, ch_t)$$

*, where $a \vee b = max(a, b)$.*

In summary, Theorem 1 helps support Definition 1. More specifically, Theorem 1 establishes a uniform high-probability bound on pairwise gap estimation error, ensuring that the empirical gap between any two arms concentrates around the true gap. This directly justifies Definition 1, which defines the "good-gap" event $\mathcal{E}$ under which all pairwise comparisons are well-behaved. Since event $\mathcal{E}$ follows from Theorem 1, Conditioned on event $\mathcal{E}$, Lemma 1 characterizes how the algorithm's adaptive confidence radii shrink over time, and Theorem 2 then converts these shrinkage properties into a high probability sample-complexity upper bound for GRASS. Overall, these results imply an upper bound on the expected number of arm pulls required by the algorithm, which translates to approximately the expected number of LLM calls needed when applying GRASS for top-*m* arms (example subsets) selection.

## 4 EXPERIMENTS

We aim to answer the following research questions: **RQ1**: Does GRASS sufficiently capture the non-linear structure of rewards ? **RQ2**: Does example selection using GRASS lead improved downstream task performance? **RQ3**: Can GRASS lead to improved task performance without sacrificing efficiency ?

---

[1] or pulling both arms in $\{b_t, c_t\}$ at time $t$

## 4.1 EXPERIMENTAL SETUP

**Datasets and Metrics:** We evaluate on diverse well-known tasks and related datasets. For numerical reasoning, we use GSM8K and AquaRAT. For demonstrating generalization abilities of our approach we also evaluate on a translation task WMT 2019, that do not require chain of thought. Detailed descriptions of the datasets are provided in Appendix F. We report performance using the official metrics: Exact Match (EM) (AquaRAT,GSM8K) and BertScore Zhang* et al. (2020) (WMT19) for the respective datasets. For **reward (LLM feedback)**, we compute BertScore Zhang* et al. (2020) between generated rationales with ground truth rationales along with generated answers for GSM8K and AquaRAT. For WMT19 we compute BertScore between generated translations **LLMs and hyperparameters:** We primarily evaluate on relatively stable open-source LLMs like Llama3.2-3b. We also report performance on closed source models like gpt-4o-mini in Appendix D and results using open models like Deepseek-R1:7B (DeepSeek-R1-Distill-Qwen-7B) are shown in Table 3. For all baselines and our approaches we set max_tokens to predict to 1000 with temperature of 0.25.

**Baselines**: We primarily compare with bandit based static example selection algorithms like CASE Purohit et al. (2025a), EXPLORA Purohit et al. (2024) and since our surrogate is based on LTR philosophy we compare with LTR baselines. **Static CASE** Purohit et al. (2025a) experiments were conducted using hyperparameters as in original work. The number of top arms to be identified was set top $m = |U_t| = 10$ and $|C_t| = 5$. The confidence parameter was fixed at $\delta = 0.05$, controlling the probability of incorrectly identifying the top-m arms. The stopping criterion which is the gap between $U_t$ and $N_t$ was also kept at $\varepsilon = 0.1$. The example subsets ($S$) are formed, by sampling *with replacement* one example from each of the 5 clusters formed from training set. We use the same hyperparameters in MAB setup for GRASS for fair comparison.

**Dynamic CASE** method is obtained by applying CASE Purohit et al. (2025a), for each test instance instead of single offline run. Moreover, the hyperparameters utilized during experiments for CASE algorithm are the same as Static CASE configuration.

**Learning to Rank baselines** - We compare to diverse LTR approaches including PiRank Swezey et al. (2021) which is based on differentiable sorting. The model architecture consisted of a sequence of fully connected hidden layers with sizes of (256,256,128,64) and ReLU activation function after each layer, processing 768-dimensional feature vectors for each document. The same hyperparameter values were also used for all LTR baselines and for non-linear surrogate in GRASS. Optimization was performed using Adam with a learning rate of $1e - 4$, paired with a StepLR scheduler with decay rate 0.1 every 50 epochs, balancing stability and convergence speed for this architecture. The training ran for 100 epochs with a batch size of 16. Parameter values for all LTR baselines are in Table 4.

## 5 RESULTS

### 5.1 EMPIRICAL VERIFICATION OF CONVERGENCE IN GRASS

To answer **RQI**, we record, compare and analyze the gap-index and simple regret across rounds. Since the stopping criterion is directly dependent on gap-index, it should decrease across rounds with minor fluctuations for convergence. The gap-index across rounds is compared across different algorithms as shown in Figure 2a. We observe that for GRASS, gap-index decreases gradually and approaches 0 demonstrating that our proposed non-linear ranking surrogate based MAB algorithm converges demonstrating it's correctness empirically. We also observe that it converges earlier than CASE. While CASE initially shows a monotonically decreasing trend in the gap, it stagnates after round 70 struggling to approach $\epsilon$. We observe that this is primarily due to the model struggling to distinguish between truly good arms and borderline challenger arms that appear to be good. We also observe simple regret as shown in Figure 2b to analyze if the estimate of good arms improves over time as the surrogate better learns to estimate the means (utility) of the arms. For GRASS, simple regret is calculated as loss of the ranked subsets in $U_t$ with respect to their true ranking based on LLM feedback based rewards. For CASE, it is calculated based on RMSE between optimal LLM reward and predicted empirical means owing to it's linear modeling of rewards. We observe that the simple regret of the set $U_t$ - the current estimate of top-m arms decreases gradually. The gap and simple regret for other datasets are reported in Appendix C

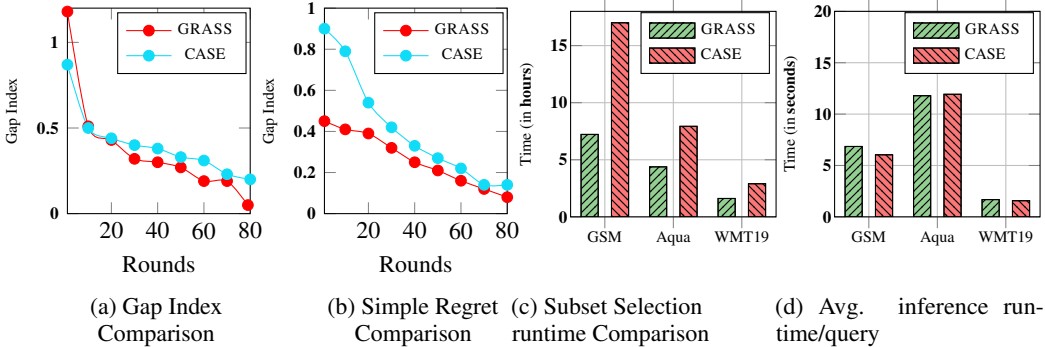

(a) Gap Index Comparison  (b) Simple Regret Comparison  (c) Subset Selection runtime Comparison  (d) Avg. inference run-time/query

Figure 2: Top-$m$ arm identification by GRASS, CASE for AquaRAT. (a) Gap Index ($B_t(s_t, b_t)$) comparison and (b) Simple regret comparison and subset selection time (c), Average inference time (d) per query across all datasets

| Method | GSM8K | AquaRat | WTM19 |
|---|---|---|---|
| **Task level** | | | |
| Zero-shot-COT Kojima et al. (2023) | 37.37 | 36.61 | 51.22 |
| Few-Shot COT Wei et al. (2023) | 63.22 | 40.94 | 61.64 |
| LENS Li & Qiu (2023) | 64.97 | 44.88 | 64.09 |
| EXPLORA Purohit et al. (2024) | 69.92 | 47.24 | 66.26 |
| Static CASE Purohit et al. (2025b) | 67.00 | 46.06 | 65.40 |
| **GRASS**$_{static}$ (ours) | 69.90 | 50.00 | 68.83 |
| **Instance Level** | | | |
| KNN Rubin et al. (2022) | 61.07 | 41.31 | 68.36 |
| MMR Ye et al. (2023b) | 66.48 | 45.84 | 68.86 |
| **Instance Level (LTR - Learning To Rank)** | | | |
| PiRank Swezey et al. (2021) | 69.30 | 37.00 | 71.30 |
| NeuralNDCG | 72.23 | 44.09 | 71.13 |
| ListNet Pobrotyn et al. (2020) | 70.30 | 43.30 | 71.26 |
| LambdaRank | 67.30 | 44.09 | 71.22 |
| NDCGLoss 2++ Wang et al. (2018) | 70.00 | 43.70 | 72.56 |
| NeuralNDCG with Normalized Data | 71.33 | 45.66 | 71.80 |
| **Instance level (Bandit Approaches)** | | | |
| $CASE_{dynamic}$ Purohit et al. (2025a) | 70.00 | 47.51 | 71.70 |
| **Instance level (Bandit + LTR)** | | | |
| **GRASS**$_{dynamic}$ (ours) | **75.66**$_{(\blacktriangle 8.08\%)}$† | **54.72**$_{(\blacktriangle 15.17\%)}$† | **78.49**$_{(\blacktriangle 9.47\%)}$† |
| **GRASS**$_{dynamic}$ (-exploration) (ablation) | 70.30 | 37.00 | 71.30 |

Table 1: Demonstration example selection results across 3 datasets using `llama3.2:3b`† indicates statistical significance (t-test) over $CASE_{dynamic}$ at 0.05 level.

## 5.2 PERFORMANCE COMPARISON FOR EXAMPLE SUBSETS SELECTION

To answer **RQ2**, we compare GRASS with static (task-level) and learning to rank based dynamic example selection approaches as shown in Table 1. Since GRASS outputs top-$m$ exemplar subsets as part of subset selection, we compare GRASS$_{static}$ with other task-level selection based inference approaches. We observe that GRASS$_{static}$ outperforms existing approaches including CASE across datasets. We hypothesize that this is primarily due to the parameterized non-linear surrogate that models the arm feature-rewards relationship better than CASE and existing approaches. We also employ the non-linear ranking surrogate trained during the selection to rank example subsets dynamically for each test instance with results indicated by GRASS$_{dyn}$ in Table 1. Firstly we observe that GRASS$_{dyn}$ outperforms static example selection approaches demonstrating need for instance-level selection as static set of examples may not generalize to unseen queries. Also from the table, we observe that GRASS$_{dyn}$ significantly outperforms existing approaches like KNN and MMR which

aim to retrieve examples based on similarity and diversity to the test example respectively. This is primarily because, $GRASS_{dyn}$ models the problem as ranking subsets as a whole than individual examples. It also takes into consideration the impact of a particular combination of examples on downstream LLM performance through the training process in the bandit optimization step. Whereas KNN and MMR retrieve examples independently without considering how they may interact together and affect LLM performance. $GRASS_{dyn}$ also outperforms dynamic version of the CASE approach which employs a bandit based selection algorithm per instance and scores subsets as a whole. For instance, $GRASS_{dyn}$ achieves upto **15.17%** over dynamic CASE on AquaRAT. We observe that the improvements are primarily because of better modeling of reward structure which also leads to clear separation between borderline and top-$m$ arms compared to CASE which employs a linear surrogate.

Comparing to LTR approaches, from Table 1, we observe that $GRASS_{dyn}$ outperforms existing LTR approaches trained with different objectives. The LTR approaches are also trained to rank example subsets as a whole than individual examples for fair comparison and demonstrate gains over static selection approaches. However, we observe that the mechanism to iteratively fit the non-linear surrogate in our approach helps the model clearly distinguish between borderline arms and top-$m$ arms through sampling of most ambiguous arms and reduction of gap between them. However, in classical LTR training approaches, there is no principled mechanism to sample ambiguous borderline arms and only adopt heuristic negative sampling without considering downstream task performance unlike $GRASS_{dyn}$ leading to sub-par performance as also evident from the ablation in Table 1.

## 5.3 EFFICIENCY COMPARISON

To answer **RQ3**, we compare the subset selection time of CASE and GRASS (Figure 2c), inference time across the three datasets. Regarding subset selection time, we observe that subset selection runtime of GRASS is less than CASE (providing approximately **2x** speedup on AquaRAT and **2.35 x** speedup on GSM8K) for identifying top-$m$ arms. We observe that this is primarily due to **faster convergence** of GRASS as per the adopted stopping criterion compared to CASE. This is because the non-linear ranking based surrogate in GRASS more accurately models the reward structure and quickly learns to distinguish between top-$m$ arms and borderline challenger arms. It accomplished this by sampling better ambiguous arms through means estimated by the surrogate. This is also evident from comparison of gap index across rounds between CASE and GRASS in Figure 2a. For instance, in AquaRAT, GRASS converges in *79 rounds*, whereas CASE requires *238 rounds*. Similarly, for GSM8k, GRASS converges in *130* rounds but CASE requires *510* rounds.

We also plot the average inference times per query of the respective methods on test set. Particularly, we compare $GRASS_{dyn}$ with static CASE to measure the latency overhead at inference time added by ranking using the non-linear surrogate in $GRASS_{dyn}$. We observe from Figure 2d, that the change in latency is negligible with $GRASS_{dyn}$ only adding to few milliseconds over static CASE. This demonstrates that $GRASS_{dyn}$ offers significant improvements over static selection methods with negligible latency overheads.

## 6 CONCLUSION

In this work, we propose a sample-efficient gap-index based MAB framework (GRASS) that models scores of example subsets using a non-linear surrogate to enhance ICL. The proposed approach provides a general mechanism to learn ranking surrogates that can learn from LLM feedback and can be employed for instance-level example subsets selection. Hence, the proposed framework can be generally be employed for task-level and instance-level selection. Since GRASS models the reward structure well it converges faster than existing MAB algorithms for subset selection in ICL setting. It adds only negligible overhead during inference for instance-level setting but with significant performance gains. The proposed algorithm can also be extended to other ranking tasks in the future with appropriate loss function changes.

## 7 REPRODUCIBILITY STATEMENT

We open source our code and related data at the anonymous github repository - `https://anonymous.4open.science/r/top-m-arm-selection-non-linear-C010`. We

have tested our algorithms on CPU and GPU till convergence. All hyperparameter details are presented in Section 4 and prompts are present in Appendix F.

## 8 LLM USAGE

We use LLMs to only correct grammatical issues.

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

## A    APPENDIX

## B    PROOF FOR THEOREM 1

**Notation and setup.**    Let $\mathcal{A}$ be the arm set, $|\mathcal{A}| = K$. At round $t$ the PiRank surrogate (with dropout) outputs $N$ stochastic predictions (N MC-dropout forward passes) per arm $a$:

$$\{y_a^{(k)}\}_{k=1}^N, \qquad \hat{\rho}_t(a) := \frac{1}{N} \sum_{k=1}^N y_a^{(k)}.$$

Define the sample mean difference and sample variance for pair $(i, j)$:

$$\bar{d}_{ij} := \frac{1}{N} \sum_{k=1}^N \left(y_i^{(k)} - y_j^{(k)}\right), \qquad \widehat{V}_{ij,t} := \frac{1}{N} \sum_{k=1}^N \left(y_i^{(k)} - y_j^{(k)} - \bar{d}_{ij}\right)^2.$$

Let $c_t$ be an annealed multiplier (user-specified) and define

$$B_t(ch_t, b_t) := \hat{\rho}_t(s) - \hat{\rho}_t(b) + W_t(b_t, ch_t).$$

In order to prove Lemma 1, we first need to demonstrate that event $\mathcal{E}$ hold with probability $\geq 1 - \delta$ for GRASS. However, the estimated means $\hat{\rho}_t$ and $W_t(i, j)$ for any two arms $i, j$ is computed in a different manner for GRASS than existing linear stochastic bandit frameworks like CASE and GIFA. Hence, we establish that the confidence event $\mathcal{E}$ holds here by deriving an upper bound on pairwise-gap error as explained below. This is one of our main **contributions** which further helps in deriving a high complexity upper bound on sample compelxity.

To recap the event is defined as, $\mathcal{E} \triangleq \bigcap_{t>0} \bigcap_{i,j \in [K]} \left(\rho_i - \rho_j \in [-B_t(j,i), B_t(i,j)]\right)$, Expanding the event $\mathcal{E}$

$$\rho_i - \rho_j \geq (\hat{\rho}_i(i) - \hat{\rho}_i(j)) - W_t(i, J)$$

and symmetrically,

$$\rho_i - \rho_j \leq (\hat{\rho}_i(i) - \hat{\rho}_i(j)) + W_t(i, j)$$

Hence it follows,

$$(\hat{\rho}_i(i) - \hat{\rho}_i(j)) - (\rho_i - \rho_j) \leq W_t(i, j)$$

Let

$$\mathbb{E}_{ij}(t) := \left(\hat{\rho}_t(i) - \hat{\rho}_t(j)\right) - \left(\rho(i) - \rho(j)\right)$$

denote the pairwise-gap error. This error can be further decomposed as follows based on the source of randomness / errors

$$\mathbb{E}_{ij}(t) = \underbrace{\left[\left(\hat{\rho}_t(i) - \hat{\rho}_t(j)\right) - \left(\mathbb{E}_{\text{drop}}[y_i] - \mathbb{E}_{\text{drop}}[y_j]\right)\right]}_{\textbf{(Term 1) MC Dropout noise}}$$

$$+ \underbrace{\left[\left(\mathbb{E}_{\text{drop}}[y_i] - \mathbb{E}_{\text{weights,drop}}[y_i]\right) - \left(\mathbb{E}_{\text{drop}}[y_j] - \mathbb{E}_{\text{weights,drop}}[y_j]\right)\right]}_{\textbf{(Term 2) Weight randomness / SGD-induced drift}}$$

$$+ \underbrace{\left[\left(\mathbb{E}_{\text{weights,drop}}[y_i] - \rho(i)\right) - \left(\mathbb{E}_{\text{weights,drop}}[y_j] - \rho(j)\right)\right]}_{\textbf{(Term 3) Model bias}}.$$

Each of the above terms can be bounded individually to prove the following theorem. Restating Theorem 1,

**Theorem (MC-Dropout gap concentration).** In a fixed-confidence setting, $\delta \in (0,1)$, with probability at least $1 - \delta$, for all pairs $i, j \in \mathcal{A}$:

$$\left|(\hat{\rho}_t(i) - \hat{\rho}_t(j)) - (\rho(i) - \rho(j))\right| \le c_t \sqrt{\frac{2\widehat{V}_{ij,t} \log(2K^2/\delta)}{N}} + \varepsilon_t^{\text{stab}} + b_i + b_j$$

$$+ \frac{2M \log(2K^2/\delta)}{3N}.$$

where $\widehat{V}_{ij,t}$ is the empirical variance of MC-dropout differences, $\varepsilon_t^{\text{stab}}$ is the SGD stability error, and $b_i, b_j$ are surrogate approximation biases.

**Term 1 - Monte-Carlo Dropout Noise**

For arms i,j we define Monte Carlo Dropout samples of their differences as:

$$Z_k = (y_i(k) - y_j(k)) - (\mathbb{E}_{\text{drop}}[y_i] - \mathbb{E}_{\text{drop}}[y_j])$$

Our goal is to bound the empirical mean $\hat{Z} = \frac{1}{N} \sum_{k=1}^{N} Z_k$ which is equivalent to Term 1.

Assuming $Z_k$'s are independent ( as dropout masks are independent) and bounded, we apply Bernstein's inequality as it states that,

For $\{Z_k\}_{k=1}^{N}$ be independent, mean-zero random variables with $|Z_k| \le b$ almost surely, and let

$$\hat{Z}_N := \frac{1}{N} \sum_{k=1}^{N} Z_k, \qquad \sigma^2 := \text{Var}(Z_k).$$

Then, for any $\epsilon > 0$, the (scalar) Bernstein inequality states: if $Z_1, \ldots, Z_N$ are independent, mean-zero, and satisfy $|Z_k| \le b$, then for any $\epsilon > 0$,

$$\Pr\left(|\bar{Z}| \ge \epsilon\right) \le 2 \exp\left(-\frac{N\epsilon^2}{2\sigma^2 + \frac{2}{3}b\epsilon}\right).$$

Here $b = 2M$ and $\sigma^2 = \text{Var}(y_i - y_j)$.

To get a confidence radius $\epsilon$ such that the event holds with probability at least $1 - \delta$ we set

$$2 \exp\left(-\frac{N\epsilon^2}{2\sigma^2 + \frac{2}{3}b\epsilon}\right) = \delta.$$

This yields the inequality:

$$2\left(-\frac{N\epsilon^2}{2\sigma^2 + \frac{2}{3}b\epsilon}\right) \ge log\delta.$$

hence,

$$\left(\frac{N\epsilon^2}{2\sigma^2 + \frac{2}{3}b\epsilon}\right) \ge log(2/\delta).$$

Following Maurer & Pontil (2009); Lattimore & Szepesvári (2020) we aim to solve the inequality:

$$\left(\frac{N\epsilon^2}{2\sigma^2 + \frac{2}{3}b\epsilon}\right) \ge log(2/\delta).$$

This can be expressed as a quadratic in $\epsilon$,

$$N\epsilon^2 - \frac{2}{3}b\epsilon(log(\frac{2}{\delta})) - 2\sigma^2 log(\frac{2}{\delta}) \ge 0$$

Solving the quadratic equation we get,

$$\epsilon \geq \frac{\frac{2}{3}b\log(2/\delta) + \sqrt{\left(\frac{2}{3}b\log(2/\delta)\right)^2 + 8N\sigma^2\log(2/\delta)}}{2N}. \tag{5}$$

To obtain an upper bound for square root term we use the inequality,

$\sqrt{a^2 + x} \leq a + \sqrt{x}$, where $a = \left(\frac{2}{3}b\log(2/\delta)\right)$ and $x = 8N\sigma^2\log(2/\delta)$

hence,

$$\sqrt{\left(\frac{2}{3}b\log(2/\delta)\right)^2 + 8N\sigma^2\log(2/\delta)} \leq \left(\frac{2}{3}b\log(2/\delta)\right) + \sqrt{8N\sigma^2\log(2/\delta)}$$

Using above in Equation 5

$$\epsilon \leq \frac{\frac{2}{3}b\log(2/\delta) + \left(\frac{2}{3}b\log(2/\delta)\right) + \sqrt{8N\sigma^2\log(2/\delta)}}{2N}$$

$$\epsilon \leq \frac{\frac{4}{3}b\log(2/\delta) + \sqrt{8N\sigma^2\log(2/\delta)}}{2N}$$

$$\epsilon \leq \sqrt{\frac{2\sigma^2\log(2/\delta)}{N}} + \frac{b\log(2/\delta)}{3N}.$$

Thus, with probability at least $1 - \delta$,

$$|\bar{Z}| \leq \sqrt{\frac{2\sigma^2\log(2/\delta)}{N}} + \frac{2M\log(2/\delta)}{3N}.$$

Replacing $\sigma^2$ by the empirical variance:

Define the sample variance estimator

$$\widehat{V}_{ij,t} := \frac{1}{N}\sum_{k=1}^{N}\left[\left(y_i^{(k)} - y_j^{(k)}\right) - \bar{d}_{ij}\right]^2, \qquad \bar{d}_{ij} := \frac{1}{N}\sum_{k=1}^{N}\left(y_i^{(k)} - y_j^{(k)}\right).$$

Then by concentration of empirical variance (again via Bernstein or Bennett bounds), $\widehat{V}_{ij,t}$ is close to $\sigma^2$ with high probability, so we may plug $\widehat{V}_{ij,t}$ into the bound:

$$|\bar{Z}| \leq \sqrt{\frac{2\widehat{V}_{ij,t}\log(2/\delta)}{N}} + \frac{2M\log(2/\delta)}{3N}.$$

**Union bound over all pairs.** We require the inequality to hold for all pairs $(i,j) \in \mathcal{A}$ simultaneously. Since there are at most $K^2$ ordered pairs, set

$$\delta = \frac{\delta}{K^2}.$$

By a union bound, with probability at least $1 - \delta$,

$$\forall i,j \in \mathcal{A}: \quad \left[\left(\hat{\rho}_t(i) - \hat{\rho}_t(j)\right) - \left(\mathbb{E}_{\text{drop}}[y_i] - \mathbb{E}_{\text{drop}}[y_j]\right)\right] \leq W_t(i,j).$$

This defines the desired pairwise confidence width $W_t(i,j)$ under Monte Carlo dropout.

**Term 2 — Weights Randomness / SGD Stability**

We need to bound

$$\left| \mathbb{E}_{\text{drop}}[y_a] - \mathbb{E}_{\text{weights, drop}}[y_a] \right|,$$

that is, the gap between the conditional dropout mean (given current weights trained on the dataset) and the expectation over randomness in the training set and weights.

To do this it is first essential that the predictions of the differentiable sorting surrogate does not deviate a lot in each round of arm sampling. This translates to proving that for one-epoch SGD the *uniform stability* criterion holds.

This criterion guarantees that a randomized algorithm is uniformly stable, if for all data sets differing in only one element, the learned models produce nearly the same predictions. This is applicable to our setup, as in each round after sampling reward from a arm, this new sample (arm+reward) is added to the training set to update the non-linear surrogate with SGD simulating a single epoch of NN training.

For one-epoch SGD, Hardt et al. (2016) show uniform stability bounds of the form

$$\sup_z \left| \ell(\text{SGD}(S), z) - \ell(\text{SGD}(S^{(i)}), z) \right| \le \varepsilon_t^{\text{stab}},$$

, where $S$ and $S^{(i)}$ differ in atmost one data sample and $\ell(\text{SGD}(S^{(i)}), z)$ denotes the loss of the randomized algorithm (Diffsort neural network). which can be translated to a bound on predictions. Under our assumptions (bounded gradients $G$ and Lipschitz loss $L$), one-epoch SGD has stability that decays with the dataset size and step size; we encapsulate this as $\varepsilon_t^{\text{stab}}$.

Concretely, there exist constants (depending on $G$, $L$, $\eta_t$) such that

$$\forall a : \quad \left| \mathbb{E}_{\text{dropout}}[y_a \mid \text{weights}] - \mathbb{E}_{\text{weights, dropout}}[y_a] \right| \le \varepsilon_t^{\text{stab}}.$$

Hence for the pair $(i, j)$, the contribution is at most $2\varepsilon_t^{\text{stab}}$; we absorb a factor of 2 into the constant and state the theorem with one $\varepsilon_t^{\text{stab}}$ representing the pairwise bound (or keep $+\varepsilon_t^{\text{stab}}$ per side — we used one in the statement for brevity).

**Intuition:** Because we train only one epoch per new sample, the model is only mildly unstable: removing or adding one sample cannot arbitrarily change predictions. That bounded change becomes a bias term in the final gap bound.

**Term 3 - Model Bias** :

The per arm bias can be defined as:

$$b_a := \left| \mathbb{E}_{\text{weights,drop}}[y_a] - \rho(a) \right|.$$

By the triangle inequality,

$$\left| \mathbb{E}_{\text{weights,drop}}[y_i] - \rho(i) \right| - \left| \mathbb{E}_{\text{weights,drop}}[y_j] - \rho(j) \right|$$
$$\le \left| \mathbb{E}_{\text{weights,drop}}[y_i] - \rho(i) \right| + \left| \mathbb{E}_{\text{weights,drop}}[y_j] - \rho(j) \right| = b_i + b_j$$

Hence,
$$\left| \mathbb{E}_{\text{weights,drop}}[y_i] - \rho(i) \right| - \left| \mathbb{E}_{\text{weights,drop}}[y_j] - \rho(j) \right| \le b_i + b_j$$

This implies that the model-bias contribution to the pairwise error is at most the sum of the two per-arm biases. If the surrogate is well-specified $b_a = 0$.

Where empirically, $b_a$ can be computed as the deviation in empirical mean of the arm over the rounds with respect to a moving average of estimated empirical means over past rounds.

Combining bounds for Term 1 ,2 and 3 yields the expression for $W_t$ in Theorem 1. The event $\mathcal{E}$ holds for this $W_t$ which is one of our **main theoretical contributions**. Then the proof for Lemma 1 follows

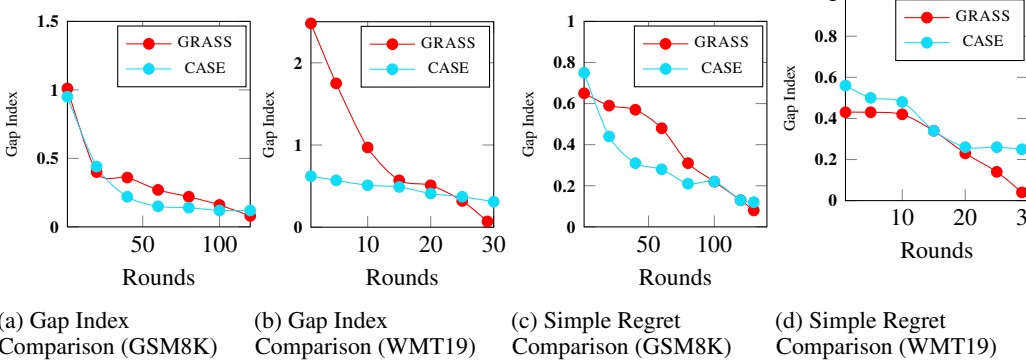

Figure 3: Top-$m$ arm identification by GRASS, CASE for GSM8k, WMT19. (a,b) Gap Index ($B_t(s_t, b_t)$) comparison and (c,d) Simple regret comparison

| Method | GSM8K | AquaRat | WMT |
|---|---|---|---|
| **Task level** | | | |
| EXPLORA Purohit et al. (2024) | 93.63 | 69.29 | 84.55 |
| LENS (Li & Qiu, 2023) | 76.19 | 64.56 | 83.57 |
| Static CASE | 91.13 | 73.23 | 83.49 |
| GRASS$_{static}$ | 94.84 | 77.16 | 86.25 |
| **Instance level** | | | |
| $CASE_{dynamic}$ Purohit et al. (2025a) | 92.19 | 76.77 | 86.36 |
| **GRASS$_{dyn}$** | 94.66 | **81.88** † | **92.41**† |

Table 2: Results across datasets (we use 5-shot for all methods) using gpt-4o-mini. † indicates statistical significance (t-test) over $CASE_{dynamic}$ at 0.05 level

from Réda et al. (2021); Purohit et al. (2025a) which yields a high probability upper bound on sample complexity as stated in Theorem 2. We include proof of Lemma 1 in Appendix E.1 for completion. We then derive the **high probability upper bound on sample complexity** In Appendix E.2.

## C  GAP AND SIMPLE REGRET ANALYSIS FOR GSM8K AND WMT19

Similar to the gap index and simple regret analysis on AquaRAT in Section 5.1, we also plot the gap indices and simple regret across rounds for GSM8K and WMT 2019 as shown in Figure 3. We observe a similar trend where gap index between ambiguous arms approaches $\epsilon$, the stopping criterion and the regret also minimizes across rounds demonstrating the empirical convergence of GRASS.

## D  RESULTS ON ALTERNATIVE LLMS

We also evaluate static and dynamic versions of our proposed approach GRASS using alternative LLMs like gpt-4o-mini and Deepseek-R1:7B (DeepSeek-R1-Distill-Qwen-7B). We choose these LLMs as it has shown relatively stable performance across benchmarks. We extract the most competitive baselines and method from Table 1 and evaluate them using gpt-4o-mini and Deepseek-R1:7B (DeepSeek-R1-Distill-Qwen-7B). All our experiments are carried out ina transfer setting where exemplars selected using Llama3.2:3b in the optimization loop are employed directly for inference on test set using gpt-4o-mini. The results using gpt-4o-mini are as shown in Table 2 and results for and Deepseek-R1:7B (DeepSeek-R1-Distill-Qwen-7B) are shown in Table 3.

| Method | GSM8K | AquaRat | WMT |
|---|---|---|---|
| **Task level** | | | |
| EXPLORA Purohit et al. (2024) | 82.63 | 68.10 | 78.59 |
| LENS (Li & Qiu, 2023) | 77.33 | 57.87 | 76.98 |
| Static CASE | 83.09 | 69.29 | 78.26 |
| GRASS$_{static}$ | 86.12 | 70.47 | 79.27 |
| **Instance level** | | | |
| $CASE_{dynamic}$ Purohit et al. (2025a) | 85.98 | 69.68 | 79.78 |
| **GRASS$_{dyn}$** | **88.40** † | **74.41** † | **81.50** |

Table 3: Results across datasets using Deepseek-R1:7b (we use 5-shot for all methods). † indicates statistical significance (t-test) over $CASE_{dynamic}$ at 0.05 level

# E    SAMPLE COMPLEXITY UPPER BOUND

## E.1    PROOF OF LEMMA 1

*Proof.* We primarily follow the proof structure of GIFA framework (Réda et al., 2021) and Purohit et al. (2025a).

**Preliminaries Recap**: Let $\mathcal{S}_m^\star$ be the true set of top-$m$ arms and $(S_m^*)^c$ denote the true set remaining worst arms. The gap-index between any two arms $i, j$ is computed as: $B_t(i, j) = \hat{\rho}_t(i) - \hat{\rho}_t(j) + W_t(i, j)$.

, where

$$W_t = c_t \sqrt{\frac{2\widehat{V}_{ij,t} \log(2K^2/\delta)}{N}} + \varepsilon_t^{\text{stab}} + b_i + b_j + \frac{2M \log(2K^2/\delta)}{3N}. \tag{6}$$

as derived in the proof for Theorem 1 To prove Lemma 1, we introduce the following property,

**Property 1**: For $b_t \in U_t$ and $ch_t \in C_t$ it holds that $\hat{\rho}_t(b_t) \geq \hat{\rho}_t(ch_t)$. Hence, it follows that $B_t(ch_t, b_t) = \hat{\Delta}_t(ch_t, b_t) + W_t(b_t, ch_t) \leq W_t(b_t, ch_t)$ as $\hat{\Delta}_t(ch_t, b_t) < 0$ From property 1, we can establish that $B_t(ch_t, b_t) \leq W_t(b_t, ch_t)$. Hence, to show that

$$B_t(ch_t, b_t) \leq -(\Delta(b_t) \vee \Delta(ch_t)) + 3W_t(b_t, ch_t)$$

we consider the following scenarios:

**(i)**    $b_t \in \mathcal{S}_m^\star$ **and** $ch_t \notin \mathcal{S}_m^\star$: In that case,

$$\Delta(b_t) = \rho(b_t) - \rho(m + 1); \Delta(ch_t) = \rho(m) - \rho(ch_t)$$

is the true gap of the arms.

As event $\mathcal{E}$ holds from Theorem 1 and Appendix B,

$$B_t(ch_t, b_t) = -B_t(b_t, ch_t) + 2W_t(b_t, ch_t) \leq \Delta(ch_t, b_t) + 2W_t(b_t, ch_t)$$

As $ch_t \notin \mathcal{S}_m^\star$,

$$\rho(ch_t) \leq \rho(m + 1)$$

$$\Delta(ch_t, b_t) \leq \rho(m + 1) - \rho(b_t) = -\Delta(b_t)$$

But as $b_t \in \mathcal{S}_m^\star$, it also holds that $\rho(b_t) \geq \rho(m)$, and $\Delta(ch_t, b_t) \leq \rho(ch_t) - \rho(m) = -\Delta(ch_t)$. Hence,

$$B_t(ch_t, b_t) \leq -(\Delta(b_t) \vee \Delta(ch_t)) + 2W_t(b_t, c_t)$$
$$\leq -(\Delta(b_t) \vee \Delta(ch_t)) + 3W_t(b_t, c_t).$$

**(ii)** $b_t \notin \mathcal{S}_m^\star$ **and** $ch_t \in \mathcal{S}_m^\star$ :

$$\Delta(ch_t) = \rho(ch_t) - \rho(m+1); \Delta(b_t) = \rho(m) - \rho(b_t)$$

By Property 1,

$$B_t(ch_t, b_t) \leq W_t(b_t, ch_t)$$
$$\leq \hat{\Delta}_t(b_t, ch_t) + W_t(b_t, ch_t) = B_t(b_t, ch_t)$$

as $\hat{\rho}_t(b_t) \geq \hat{\rho}_t(ch_t)$. Further, as $\mathcal{E}$ holds,

$$B_t(b_t, ch_t) = -B_t(ch_t, b_t) + 2W_t(b_t, ch_t)$$
$$\leq \Delta(b_t, ch_t) + 2W_t(b_t, ch_t)$$

As $b_t \notin \mathcal{S}_m^\star$, $\rho(b_t) \leq \rho(m+1)$ and hence $\Delta(b_t, ch_t) \leq \rho(m+1) - \rho(ch_t) = -\Delta(ch_t)$ As $ch_t \in \mathcal{S}_m^\star$, $\rho(ch_t) \geq \rho(m)$ and hence $\Delta(b_t, ch_t) \leq \rho(b_t) - \rho(m) = -\Delta(b_t)$. Hence,

$$B_t(ch_t, b_t) \leq -(\Delta(b_t) \vee \Delta(ch_t)) + 2W_t(b_t, c_t)$$
$$\leq -(\Delta(b_t) \vee \Delta(ch_t)) + 3W_t(b_t, c_t).$$

**(iii)** $b_t \notin \mathcal{S}_m^\star$ **and** $ch_t \notin \mathcal{S}_m^\star$: We state that there exists a $b \in \mathcal{S}_m^\star$ that belongs to $C_t$. At any time t,

$$M_t \leftarrow random\ m'arms\ from\ (U_t \cup N_{t-1})^c$$

$$C_t \leftarrow \text{top}_{m'}(M_t \cup N_{t-1}; \hat{\rho}_{(t-1)})$$

Due to the above sampling approach adopted for $C_t$ which captures the next m' arms with the highest means, we posit that $C_t$ captures at least one arm in $\mathcal{S}_m^\star$. Assuming the event $\mathcal{E}$ holds and $b \in \mathcal{S}_m^\star$,

$$W_t(b_t, ch_t) \geq B_t(ch_t, b_t) \geq B_t(b, b_t)$$

$ch_t$ by the definition is one of the most ambiguous arms posing largest threat to $b_t$ as it has the largest gap with respect to $b_t$ $B_t(ch_t, b_t) \geq B_t(b, b_t)$. Hence, $B_t(ch_t, b_t) \geq B_t(b, b_t)$. From this and event $\mathcal{E}$ it follows

$$B_t(ch_t, b_t) \geq B_t(b, b_t) \geq \rho(b) - \rho(b_t) \geq \rho(m) - \rho(b_t)$$

. Hence $W_t(b_t, ch_t) \geq B_t(ch_t, b_t) \geq \Delta(b_t)$. Using event $\mathcal{E}$,

$$B_t(ch_t, b_t) \leq \Delta(ch_t, b_t) + 2W_t(b_t, ch_t) = (\rho(ch_t) - \rho(m)) +$$
$$(\rho(m) - \rho(b_t)) + 2W_t(b_t, ch_t)$$

From above Eq and since $B_t(ch_t, b_t) \geq \Delta(b_t)$,

$$B_t(ch_t, b_t) \leq -\Delta(ch_t) + \Delta(b_t) + 2W_t(b_t, ch_t)$$
$$\leq -\Delta(ch_t) + 3W_t(b_t, ch_t)$$

Also from Property 1 and $W_t(b_t, ch_t) \geq \Delta(b_t)$, it holds that

$$B_t(ch_t, b_t) \leq W_t(b_t, ch_t) = -W_t(b_t, ch_t) + 2W_t(b_t, ch_t)$$
$$\leq -\Delta(b_t) + 2W_t(b_t, ch_t) \leq -\Delta(b_t) + 3W_t(b_t, ch_t)$$

Hence $B_t(ch_t, b_t) \leq -(\Delta(b_t) \vee \Delta(ch_t)) + 3W_t(b_t, c_t)$.

**(iv)** $b_t \in \mathcal{S}_m^\star$ **and** $ch_t \in \mathcal{S}_m^\star$: Then there exists a $s \notin S_m^*$ and $s \in U_t$ In that case,

$$\Delta(b_t) = \rho(b_t) - \rho(m+1); \Delta(ch_t) = \rho(ch_t) - \rho(m+1)$$

Also by definition of $b_t$ and $ch_t$, it holds that $B_t(ch_t, b_t) = \max_{i \in U_t} \max_{j \in C_t} [B_t(j, i)]$ Since there exists $s \in U_t$ and $ch_t \in C_t$,

$$B_t(ch_t, b_t) = \max_{i \in U_t} \max_{j \in C_t} [B_t(j, i)] \geq \max_{j \in C_t} B_t(j, s)$$
$$\geq B_t(ch_t, s) \geq \rho(ch_t) - \rho(s) \geq \rho(ch_t) - \rho(m+1)$$

As $\rho(ch_t) - \rho(m+1) = \Delta(ch_t)$, $B_t(ch_t, b_t) \geq \Delta(ch_t)$ By property 1, $B_t(ch_t, b_t) \leq W_t(b_t, ch_t)$. Hence,

$$\Delta(ch_t) \leq B_t(ch_t, b_t) \leq W_t(b_t, ch_t)$$

On event $\mathcal{E}$ it follows that $B_t(ch_t, b_t) \leq \rho(ch_t) - \rho(b_t) + 2W_t(b_t, ch_t)$ as $(B(ch_t, b_t) \leq W_t(b_t, ch_t)$. Then $\rho(ch_t) - \rho(b_t)$ can be expressed as $\rho(ch_t) - \rho(m+1) + \rho(m+1) - \rho(b_t)$. hence,

$$B_t(ch_t, b_t) \leq \rho(ch_t) - \rho(m+1) + \rho(m+1) - \rho(b_t)$$
$$+2W_t(b_t, ch_t) \leq \Delta(ch_t) - \Delta(b_t) + 2W_t(b_t, ch_t)$$

We already know that $B_t(ch_t, b_t) \geq \Delta(ch_t)$ resulting in,

$$(a) \ B_t(ch_t, b_t) \leq -\Delta(b_t) + 3W_t(b_t, ch_t)$$

Now to prove $B_t(ch_t, b_t) \leq -\Delta(ch_t) + 3W_t(b_t, ch_t)$, we rely on property 1,

$$B(ch_t, b_t) \leq W_t(b_t, ch_t) \leq -W_t(b_t, ch_t) + 2W_t(b_t, ch_t)$$

As $W_t(b_t, ch_t) \geq \Delta(ch_t)$, $-W_t(b_t, ch_t) \leq -\Delta(ch_t)$. Hence,

$$(b) \ B(ch_t, b_t) \leq W_t(b_t, ch_t) \leq -W_t(b_t, ch_t) + 2W_t(b_t, ch_t)$$
$$\leq -\Delta(ch_t) + W_t(b_t, ch_t) \leq -\Delta(ch_t) + 3W_t(b_t, ch_t)$$

From (a) and (b)

$$B_t(ch_t, b_t) \leq -(\Delta(b_t) \vee \Delta(ch_t)) + 3W_t(b_t, c_t) \tag{7}$$

$\square$

## E.2    PROOF BLUEPRINT FOR THEOREM 2

*Proof.* We now convert equation 7 into sampling bounds by using the stopping rule and the explicit form of $W_t$. The intuition is similar to Lemma 8 in GIFA Réda et al. (2021), where once the stopping rule $B_t(ch_t, b_t) \leq \varepsilon$ triggers, arms with non-zero gap must have been sampled enough times so that the width is small relative to the gap. We invert this relation to obtain a per-arm bound.

**Stopping rule.**    Assume the algorithm stops when

$$B_t(ch_t, b_t) \leq \varepsilon.$$

On the event $\mathcal{E}$, by Lemma 1 at time t < stopping time we have

$$\varepsilon \leq B_t(ch_t, b_t) \leq -(\Delta(b_t) \vee \Delta(ch_t)) + 3W_t(b_t, ch_t).$$

Rearrange to get

$$3W_t(b_t, ch_t) \geq \varepsilon + (\Delta(b_t) \vee \Delta(ch_t)).$$

$$W_t(b_t, ch_t) \geq \frac{\epsilon + \Delta_a}{3}$$

Hence, for any arm $a$ that remains active (i.e. is sampled further until elimination), when it is sampled at time $t$ its associated width at that time must satisfy the above inequality (with $a$ playing the role of $b_t$ or $ch_t$ in the identity). Substituting leading term of $W_t(i, j)$ (first term in Equation 6): $c_t \sqrt{\frac{2\widehat{V}_{b_t,ch_t,t} \, log(2K^2/\delta)}{N}}$ (we ignore stability and bias terms for clarity and also because they are negligible)

$$c_t \sqrt{\frac{2\widehat{V}_{b_t,ch_t,t} \, log(2K^2/\delta)}{N}} \geq \frac{\epsilon + \Delta_a}{3}$$

Since the predictive variance decreases with the number of arm samples $n_t(a)$

$$\widehat{V}_{b_t,ch_t,t} \leq \frac{\sigma_{a,t}^2}{n_t(a)}$$

Where $\sigma_{a,t}^2$ is the effective variance of arm $a$

Substituting this in the inequality from earlier we get,

$$c_t \sigma_{a,t} \sqrt{\frac{2 \, log(2K^2/\delta)}{N.n_t(a)}} \geq \frac{\epsilon + \Delta_a}{3}$$

Taking square on both sides

$$c_t^2 \sigma_{a,t}^2 \frac{2 \, log(2K^2/\delta)}{N.n_t(a)} \geq \frac{(\epsilon + \Delta_a)^2}{9}$$

$$n_t(a) \leq 18 c_t^2 \sigma_{a,t}^2 \frac{log(2K^2/\delta)}{N(\epsilon + \Delta_a)^2}$$

To account for case when $\Delta_a$ is tiny we replace $(\epsilon + \Delta_a)^{-2}$ yielding

$$n_t(a) \leq 18 c_t^2 \sigma_{a,t}^2 \frac{log(2K^2/\delta)}{N} . \max\left\{\varepsilon^{-2}, \, \left(\tfrac{\varepsilon + \Delta_a}{3}\right)^{-2}\right\}$$

**Per-arm and total bounds.** Formally, on event $\mathcal{E}$, for every arm $a$,

$$\mathcal{N}_T(a) \leq 18 c_t^2 \sigma_{a,t}^2 \frac{log(2K^2/\delta)}{N} \cdot \max\left\{\varepsilon^{-2}, \, \left(\tfrac{\varepsilon + \Delta(a)}{3}\right)^{-2}\right\}. \tag{8}$$

Summing over arms yields the total-sample upper bound

$$T \leq 18 c_t^2 \frac{log(2K^2/\delta)}{N} \sum_{a \in \mathcal{A}} \sigma_{a,t}^2 \cdot \max\left\{\varepsilon^{-2}, \, \left(\tfrac{\varepsilon + \Delta(a)}{3}\right)^{-2}\right\}. \tag{9}$$

$\square$

The above equation leads to the upper bound on sample complexity as stated in Theorem 2.

$\square$

# F DATASET DESCRIPTION

: **AquaRAT**: It comprises 100,000 algebraic word problems in the train set with dev and test set each comprising 254 problems. The problems are provided along with answers and rationales providing the step-by-step solution to the problem.

**GSM8K**: This dataset consists of linguistically diverse math problems that require multi-step reasoning. The dataset consists of 8.5K problems and we evaluate on the test set of 1319 questions.

**WMT 19**: We focus on en-zh (english-chinese) translation split. Test sets are a few thousand sentences (for example, 3,981 in WMT18 for zh-en direction for test). Train set has 173k english-chinese sentence pairs.

# G LTR HYPERPARAMETERS

The hyperparameters for learning to rank baselines are detailed in Table 4.

Table 4: Details of hyperparameters used in different LTR model configurations. Categorized by loss function and framework.

| Framework | Loss function | Hyperparameters | Network architecture |
|---|---|---|---|
| **PiRank** | PiRank surrogate loss | Presented in Section 4 | (256, 256,128,64) |
| Other LTR baselines | Neural NDCG | N = 2, $d_{ff} = 384$, h = 1, dropout = 0.1 | (768,96) |
| | ListNet | N = 4, $d_{ff} = 512$, h = 2, dropout = 0.3 | (768,128) |
| | LambdaRank | N = 2, $d_{ff} = 384$, h = 1, dropout = 0.1 | (768,96) |
| | Neural NDCG | N = 4, $d_{ff} = 512$, h = 4, dropout = 0.3 | (768,96) |
| | Neural NDCG With Normalized data | N = 2, $d_{ff} = 384$, h = 1, dropout = 0.1 | (768,96) |
| | NDCGLoss 2++ | - | (256, 512, 1024, 512, 256) |

# H DATASET PROMPTS

The prompts are given in Figures 4, 5. The prompts for WMT19 are in the anonymous github repo.

**AQUA Prompt**

**Instruction**:You are a helpful, respectful, and honest assistant helping to solve math word problems or tasks requiring reasoning or math. Follow given examples and solve the problems in step by step manner.

**Exemplars** :
[Question]: *The average age of three boys is 45 years and their ages are in proportion 3:5:7. What is the age in years of the youngest boy?*
[Options]: A) 9, B) 10, C) 11, D) 12, E) 13
[Explanation]: $3x + 5x + 7x = 45$,
$x = 3$,
$3x = 9$
[Answer]: The option is A
$\cdots$
$\cdots$

**Test Input** : Question: Options:
Explanation: [INS] Answer: [INS]

Figure 4: Prompt for Aqua

**GSM8K Prompt**

**Instruction**:You are a helpful, respectful and honest assistant helping to solve math word problems or tasks requiring reasoning or math. Follow given examples and solve the problems in step by step manner.

**Exemplars** :
[Question]: *Samir just turned half the age Hania was 10 years ago. If in five years Hania will be 45 years old, what will Samir's age be five years from now?*
[Explanation]: If in five years, Hania will be 45 years old, currently she is $45 - 5 = 40$ years old. Samir just turned half the age Hania was 10 years ago, which means she is $30/2 = 15$ years old. In five years, Samir will be $15 + 5 = 20$ years old.
[Answer]: 20 years old
$\cdots$
$\cdots$

**Test Input** : Question:
Explanation: [INS] Answer: [INS]

Figure 5: Prompt for GSM8K

