# OpenReview forum: "A Non-Linear Ranking Surrogate based Stochastic Bandits for top-m arm Selection"
_ICLR.cc/2026/Conference — Submitted to ICLR 2026_

### Official Review · Reviewer_HvfP · 2025-10-31

**Soundness:** 1
**Presentation:** 2
**Contribution:** 1
**Rating:** 2
**Confidence:** 4

**Summary:**

This paper considers the problem of exemplar selection for in-context learning (ICL) for large language models (LLMs). Authors formulate it as a top-m arm identification problem in a multi-armed bandit framework. Importantly, an arm in their setting is something complex: it is a set of exemplars for the prompt of LLM. The authors found that existing approaches assume linear reward relationships, which are insufficient for capturing complex, non-linear dependencies of the answer of a neural network (LLM) on the characteristics of the exemplars. To overcome this, they develop GRASS, a novel gap-index bandit method with a non-linear ranking surrogate trained offline.

**Strengths:**

TBD: strong point will be discussed when the conceptual problem on the formulation is resolved.

**Weaknesses:**

I see a deep conceptual drawback in the problem formulation of this paper. In a nutshell, the formulation follows a previous work, “Sample Efficient Demonstration Selection for In-Context Learning” by Purohit et al. Unfortunately, I need to continue with a closer examination of that previous paper, since there are some tricky points I cannot grasp unless the authors broaden my understanding. In the Problem Setup section (3.1), the initial problem is correctly formulated as  $\arg \max_U$ … (Equation 2), where $U$ is a set of $m$ arms (sets), which (in my mind) should COVER the diversity of the validation set of examples. However, immediately in the next subsection, this problem somehow transforms to a top-m arm selection, where the problem is already $\arg \max_a$, with $a$ being a single arm, which should therefore cover the whole validation set  ON AVERAGE.

Could you elaborate on the connection between the initial (correct) problem in Section 3.1 and Section 3.2 of the paper by Purohit et al.?

To be honest, there are some inaccurate statements in the paper by Purohit et al. which I cannot agree, so I cannot draw conclusions based on their results. For example, they argue in section 3.2 that “the number of arms is exponential in both k and n.” It is surely not true. The number of arms is  $n!/(k! (n-k)!)$, what is exponential in k, and polynomial in n.

**Questions:**

See Weaknesses above

---

> ### Author Response · Authors · 2025-11-12
> **Response to Reviewer HvfP**
>
> We thank the reviewer for their time and feedback. We would like to clarify some of the queries regarding interpretation of work [1].
>
> Q1. **The formulation follows prior work [1]. There are some tricky points I cannot grasp unless the authors broaden my understanding. In the section 3.1,  (Eq 2), which (in my mind) should COVER the diversity of the validation set of examples. However, in 3.2, this problem transforms to a top-m selection, where a single arm, should cover the whole validation set ON AVERAGE.**
>
> **Response**: We would like to clarify the interpretation. Equation 2 *does not solely imply* each arm in U* will cover only some parts of validation set and hence implying  U* as a whole covers diversity of samples in validation set.
>
> As per definition of U* in Section 3.1 penultimate paragraph on page 3 of [1]   the goal is static exemplar selection, that is, to find U* , **the highest scoring exemplar subsets**. The objective is framed in this manner because multiple exemplar subsets could lead to similar high performance on average & goal is to identify & **separate near-optimal arms** from **less-than-optimal/borderline arms (exemplar subsets)** leading to the top-m formulation inspired by Reda et. al as mentioned in [1], where the goal is to identify top-m arms with highest means. Hence, equation 2 of [1] states that the authors desire high scoring top-m subsets, where an exemplar subset is selected from this top-m for each val. query by **prompt constructor $\pi$** ( it could be *same subset or different for each query* as detailed in last para. below). This implies each subset in the top-m should have higher val. set accuracy(averaged across queries in val set) than remaining subsets $\notin U^*$. This implies each subset in top-m *covers most of validation set*.
>
> Of course, there could be small relative differences in estimated means (validation set avg. accuracy) among top-$m$ arms. This implies some arms in top-m maybe better for some queries in the task than others, but each have higher means (avg. accuracy across queries in val. set) than arms not in top-m. Hence each arm in top-m should *cover most queries in validation set* on average compared to sub-optimal arms. This implies each arm in top-$m$ are on average much better than any other arm not in top-$m$. In summary, one would *want to avoid bad arms* (less-than-optimal exemplar subsets) by identifying & clearly separating arms with highest means (avg. val. acc.) which lead to better performance, which is the core of top-$m$ arm identification problem.
>
> Please note in their formulation, $\pi$ selects one subset from top-m for test query dependent selection & due to  context length limit. Hence, all top-m subsets are not provided as part of the prompt at test time. As stated by authors of [1] $\pi$ could simply select the subset from top-m with highest validation accuracy when compared to other subsets in top-m, or could use KNN where subset whose examples have highest average similarity to the test query are selected. Alternatively, $\pi$ could use MMR where the subset which has examples with more diversity relative to test query is selected.
>
>
> Q2. **Could you elaborate on the connection between  Section 3.1 & 3.2 of [1]?**
>
> **Response**:
> The formulation in [1] (Sec. 3.2) defines ICL exemplar selection as a top-$m$ arm identification task following directly from problem setup (*Section 3.1*) as detailed in previous answer. [1] instantiates this within a linear stochastic bandit framework: each exemplar subset is an arm, LLM feedback provides noisy rewards, and the gap-index method ensures sample-efficient exploration by prioritizing arms for  sampling LLM feedback. The linear surrogate $\hat{\rho}(a,\alpha)$ models (approximates) arm feature–reward relationships and is updated adaptively using the  to identify the top-$m$ subsets, while Sec. 3.3 of [1] enhances efficiency via challenger shortlists. The top-m subsets are used at test time by $\pi$ as described in previous answer.
>
> **Q3. For example, they ([1]) argue in section 3.2 that “the no. of arms is exponential in k and n.” It is surely not true.**
>
> **Response**: We would like to clarify that **Section 3.2** in [1] tries to provide the intuition of why naive, exhaustive or heuristic search would not scale for the MESS problem in Section 3.1. In theory, enumerating all possible subsets leads to exponential blow-up as k can also approach $O(n)$ (when enumerating subsets for different values of $k$), the authors state it leads to no. of arms being exponential in k & n. This shows that even in practice even for moderate k & n,exhaustive search is prohibitive.
>
>
> We hope the above answers help clarify the work ``Sample Efficient Demonstration Selection for In-Context Learning” by Purohit et al. [1]. Thank you for your time. We look forward to answering any other queries and further discussions.
>
> [1] Sample Efficient Demonstration Selection for In-Context Learning. Purohit et. al.

---

> ### Comment · Reviewer_HvfP · 2025-11-20
> **Do not agree in deep, response is INADEQUATE**
>
> Unfortunately, I cannot agree with your answer.
>
> On one hand, the motivations behind multiple subsets is clear.
>
> In your paper:
> “The quality of the chosen subset 𝑆 has a direct impact on LLM performance. However, a single 𝑘-subset is rarely sufficient: different subsets capture different reasoning skills or topical knowledge required for a query at test-time, and multiple subsets may yield comparable outcomes (on average across test queries).”
>
> In the paper “Sample Efficient Demonstration Selection for In-Context Learning”: “While the above-described basic mechanism can be used for in-context learning, a single subset of k-exemplars often may not capture all the diverse aspects of a given task. Hence, prompt generators π are used to generate prompts specific to a given test input”.
>
> On the other hand, the following your arguments contradict the motivation:
> “that implies each subset in the top-m should have higher val. set accuracy(averaged across queries in val set) than remaining subsets . This implies each subset in top-m covers most of validation set.”
>
>
> DOES NOT imply. You can just consider any toy construction theoretically to be sure I am right. For example, let the target function for in-context modeling is a piecewise linear (k-1)-dimensional function with k pieces and (k-1)-dimensional feature vector. Let the parameters of different pieces be random and independent. Let k=m. Then, obviously, the optimal m=k subsets in terms of Section 3.1 [1] is k groups of k points, where each group represents the corresponding piece / domain (so that different “arms” cover different domains and enable to fully reconstruct the corresponding linear function). However, a top arm in terms of Section 3.2 should be a set of k points representing different domains. As we can see, the structure of such top arms is deeply different from that optimal set above, and no of these top arms can provide more than only one relevant point for any test example.
>
> For me, this reasoning represents the basic intuition of dynamic selection. Doesn't it?
> If so, why do you persistently violate it?
>
> At the moment, I need to further reduce the scores.

---

> ### Author Response · Authors · 2025-11-20
> **Response to Reviewer HvfP - Clarifying interpretation of objective of [1]; central objective in GRASS (current paper) and algorithm is different from [1],  request to consider claims & algorithm, optimization differences made in current work GRASS**
>
> We believe our response has been misunderstood, we wanted to request the opportunity to clarify. We also would like to point that CASE [1]  performs **static (task-level [2,3]) selection and not dynamic selection and works differently from our current work (GRASS) as detailed later below**. **The subsets in [1] for this reason (task-level) are **not each intended to cover solely just one domain as focus is on task-level selection** and are expected to have higher average performance across queries [2,3]. The objective in equation 2 (3.1) reflects task-level selection [2,3]. Note that $pi$ selects only one subset for all test /val queries in static case and one subset per test / val query in static-dynamic case. Also bandit run for top-m selection in [1] is run once offline (non test-time).
>
> However, CASE [1]   acknowledges that there is **not only one subset of exemplars** with higher performance at task-level,  and hence frames the goal as to identify those near optimal subsets and separate them from sub-optimal exemplar sets.  Please note that our rebuttal stating "most of validation set" meant to imply these subsets provide higher coverage of the validation queries relatively when compared to sub-optimal sets which work well only for a subset of the validation queries. Please note that this is also covered by equation 2 in [1] where goal is stated as " finding m-subsets of X such that the prompt P generated by $\pi$ maximizes the total validation accuracy".  Note that  the prompt constructor selects one subset from  top-m based on lowest validation loss (static) to perform inference on all test samples,  which implies all subsets in top-m should be relatively better than other m subsets of the search space. We agree that this doesn't guarantee that each subset performs well on most of the validation set, but equation 2 (static) implies that the top-m  subsets identified achieve higher coverage than other sets which are sub-optimal due to their relatively lower coverage.
>
> Note that CASE offers a **static-dynamic hybrid option (not purely dynamic)** in form of CASE+KNN and CASE+MMR where the prompt constructor $\pi$ could specialize the subset selected for each test sample by selecting a subset of examples (only one) from top-m based on average similarity or diversity to test query as in [1] and our previous response. This helps select the best subset among top-m for each test query further improving performance as shown in their ablations. **Note there is utility here in performing dynamic selection over set of good candidate arms with higher means (relatively better average val set performance)**.
> **Please note the $\pi$ in Equation 2  for CASE[1] in main results table is the static version** and is static-dynamic hybrid for CASE+MMR & CASE+ KNN.  **Authors of  CASE should have denoted this hybrid with separate notation than $\pi$ clearly.** Hence, the goal of the offline (non test-time) bandit run is to separate sub-optimal subsets.
>
> **Please note that this is the primary objective of optimization - to uniquely identify exemplar subsets(arms) with higher relative performance (means)  by separating sub-optimal exemplar subsets**. It is also valuable for hybrid setting because pruning sub-optimal arms improves dynamic selection methods (KNN,MMR) adopted by $\pi$ (hybrid mode).
>
>  Achieving ability to rank subsets to tell apart sub-optimal arms is stated as central objective in our work GRASS (Lines **69-72** & abstract). **We frame it as a learning to rank problem**. In learning to rank tasks a primary objective is to clearly separate boundary items and identify uniquely top-m items. Hence, the  formulation which employs gap index where ambiguous arms sampled to uniquely identify top-m arms and reduce uncertainty of borderline arms by reducing the gap works well for this setting. Since, the non-linear surrogate is trained on data from sampled arms  it learns to clearly separate borderline arms from optimal arms scored w.r.t each validation sample.
>
>      **Please Note:  CASE focuses on static selection and hence uses average accuracy as reward to update the surrogate but GRASS computes a reward vector - rewards for subsets across each validation sample, mean vector and surrogate is updated based on RANKING LOSS calculated based on ranking of subsets for each validation sample as detailed in following responses aligning with dynamic selection goals**
>
> ## Please note Lines 227-231 in our paper point out the limitations in static formulation and motivate our approach detailed in 3.2
>
> *We have stated goal of our approach as learning to distinguish borderline subsets (**69-72**, abstract,**488**)* to fit the diff. sorting surrogate-employed as ranker at inference time
>
> [1]  Sample Efficient Demonstration Selection for In-Context Learning
> [2] Selecting Demonstrations for Many-Shot In-Context Learning
> [3] LENS: A Learnable Evaluation Metric for Text Simplification

---

> ### Author Response · Authors · 2025-11-20
> **Request to Reviewer HvfP  reconsider our response, consideration of claims in GRASS, **Primary differences in algorithm and formulation between GRASS and CASE****
>
> We would like to also clarify, in continuation to above, that the motivation in $GRASS_{dynamic}$ is to learn a non-linear ranking model for dynamic selection. The model is trained in a sample-efficient manner offline using a novel gap-index bandit framework that accounts for non-linearity of surrogate. Note that the bandit algorithm through gap-index based bandit formulation helps sample a training dataset for the non-linear surrogate where it implicitly learns to separate optimal arms from sub-optimal arms *using rewards (reward vector) w.r.t each val.sample as detailed in 3rd point below* & hence the algorithm is fundamental different from [1] and not a mere extension. Hence our work does not violate the dynamic selection criterion as the ranker must be able to tell apart good items from confounders with respect to specific val / test queries which is the core of any ranking task.
>
> Please note that CASE and GRASS primarily differs here though they employ the gap-index framework. **GRASS is NOT MERELY EXTENSION OF CASE with non-linear surrogate**
> 1.  GRASS primarily focuses on dynamic selection, and hence the goal is to *train a ranker in sample efficient manner that can clearly tell apart near optimal subsets from sub-optimal ones*. Hence, the top-m formulation of linear stochastic bandits serves as inspiration here. Please note the algorithm 1 trains the non-linear surrogate adaptively to separate  based on arms pulled (LLM based rewards sampled)
> 2. Except, GRASS provides new algorithmic and theoretical contributions to account for the non-linear surrogate as the linearity assumption is violated and closed form confidence sets no longer are applicable. We demonstrate it is still possible to derive data dependent upper bounds on sample complexity, serving as contributions to stochastic bandits literature.
>
> 3. Please note that  each subset is scored with respect to a validation example and rewards are sample for (val. example, subset) pairs making the reward a vector of scores. Likewise in equation 2 (**of our work not [1]**) estimated mean for an arm is with respect to each validation sample (**a primary difference between GRASS and CASE** as in CASE estimated mean is a point estimate calculated by as product of  subset parameters vector and similarity vector). This is primarily because we use a ranking loss as detailed in **Lines 309-313** is w.r.t mean & reward vectors for each val. sample.  Hence, surrogate is updated w.r.t scores assigned to subsets with respect to each validation sample aligning with needs of dynamic subset selection. We employ the top-m formulation as it ensures sufficient number of ambiguous arms are pulled which helps the ranker learn to distinguish between good subsets and borderline / sub-optimal subsets & for sample-efficiency
>
> We **request the reviewer to consider contributions of the current work GRASS, the claims made and the theoretical contributions**. We have tried our best to clarify and distinguish the prior work CASE and our work GRASS.  We *would like to point to the corresponding claims and results in GRASS Lines 23, 69-73, 494, 488-489, Section 5.3* , algorithm with instance-level (validation sample) rewards for subsets , loss , surrogate updates clearly align with the dynamic selection motivation for $GRASS_{dynamic}$. We are happy to provide any further clarifications.
>
>  Also in regard to [1] some example subsets may perform well on some queries than others because while different subsets may have similar average performance which is relatively higher (more coverage) then sub-optimal subsets, **the subset of test queries they perform well on need not be exactly similar**.  For instance, out of 100 queries, subset A could perform well on 60 samples and subset B could also perform well on similar number of samples but on a set of 60 samples where 40 could be disjoint compared to samples covered by subset A.
>
> Also,  with regards to [1] selecting top-m subsets with the highest means ensures optimal subsets from the whole search space helps both static subset based inference  and dynamic inference. This is because removal of sub-optimal arms ensures selecting any single subset from this top-m would achieve relatively better performance compared to rest of search space. Additionally, it reduces the search space for dynamic approaches like KNN and MMR paving way for static-dynamic hybrid approaches adopted in [1].
>
> **We urge and humbly request the reviewer to consider the claims made in GRASS, difference with respect to CASE (note 3.2 in our paper is the main GRASS formulation leading upto the algorithm)** with corresponding results and clarification of subsets (**Lines 23, 69-73, 488-489, Section 5.3**) which clearly align. We also believe the statement made in our paper regarding subsets with comparable performance has been misinterpreted, as we clarified in the paragraphs above.
>
> [1] Sample Efficient Demonstration Selection for In-Context Learning. Purohit et. al

---

> ### Author Response · Authors · 2025-11-21
> **Further clarifications regarding ``Sample Efficient Demonstration Selection for In-Context Learning" from public original  authors comments in response to Reviewer HvfP**
>
> We are also attaching further points to clarify interpretation of prior work CASE[1] based on publicly available comments by authors of the paper. We would also again like to highlight that optimization in our current paper GRASS is primarily to train the ranker for purely dynamic approach to rank subsets at inference  and is different from static (task-level) and static-dynamic approaches of CASE [1] in calculation of mean estimates, reward sampling, loss objective and surrogate update mechanisms. We employ gap-index based approach to primarily sample arms that help the ranker clearly distinguish good arms from sub-optimal ones.  That being said, we proceed with clarification of [1] based on the authors publicly available comments:
>
> 1. Authors of CASE [1]  assume that the top-m
>  exemplar subsets (prompts), which have the highest average accuracy (equivalent to the subsets with the highest individual accuracy), are also those that maximize the objective function in Equation (2).
>
> 2. ``Please note that in [1] the transformation $\pi$ calculates the best subset among the top-m subsets for a given validation example (static-dynamic hybrid approach as described earlier). For the total accuracy of the prompt generated by $\pi$ (Equation 2) to be high, each subset should maximize the accuracy across most validation examples in $\mathcal{V}$. The top-m objective function assumes that the exemplar subset also performs well on other validation examples, so that the average accuracy of each subset over the considered validation subset is the highest. This assumption inherently filters out subsets that are highly accurate for a few validation examples but highly inaccurate for the majority (i.e. helps filter out sub-optimal arms / subsets).
>
> This assumption is critical due to the complexity involved in the search process. The search space is large - first over all possible sets of exemplars to form subsets, and then over all subsets to identify the top-m
>  subsets. This complexity is discussed in Section 3.2"
>
> [1] Sample Efficient Demonstration Selection for In-Context Learning
>
> We hope this helps clarify [1]. We politely request the reviewer to reconsider our work in light of this clarification and also politely request to reconsider the scores. Thank you for your time and consideration.

---

> ### Author Response · Authors · 2025-11-21
> **Summary and request to Reviewer HvfP for consideration of our work**
>
> ## We believe our contributions are being misinterpreted as just a non-linear extension of [1]. While we build upon gap-index framework of stochastic bandits, our algorithm is fundamentally different from CASE as [1] focuses on task-level (static) selection and our loss, updates are tailored for ranking subsets at instance level.  **No assumptions reg. dynamic selection are violated**
>
> **Prior work clarifications**:
>  The motivation of [1] is also clarified  in our above response based on public comments from authors of CASE.  Please note that CASE[1] primarily focuses on task-level or static selection, where the goal is to find subsets of exemplars that perform comparably well with higher avg. accuracy than sub-optimal exemplars. Similar definitions can be found in prior work [2]. Equation (2) reflects this task-level selection objective (here $pi$ does not perform static-dynamic hybrid but static selection during inference). Note that CASE[1] also offers a static-dynamic hybrid where when $\pi$ , the prompt constructor, uses KNN or MMR, it applies them over good candidates (top-m). We acknowledge different notations could have been used by the authors of [1] for $\pi$ when it statically selects one subset from top-m  compared to the hybrid version to avoid confusion. Since, authors want to reduce chances of selecting a sub-optimal subsets, pruning them using the top-m formulation is beneficial.
>
> Also, please note the example given in previous comments, where subsets with similar average performance could differ in the test /val queries they cover. This is what the ``diverse aspects" refers to in [1]  Hence, the authors of CASE [1] also propose a static-dynamic hybrid inference approach (CASE+MMR, CASE+KNN), where they employ dynamic selection over the selected top-m subsets. Note, CASE[1] is primarily task-level selection where definition of having high quality exemplars that have higher average test/val set performance also follows from prior works [2]. We believe this clarifies that primary objective of CASE is not dynamic selection but task-level (static) & also offers extension to static-dynamic hybrid at inference time.
>
> **Our current work's Contributions**
>
> Also, **the objective of gap-index framework in our paper (GRASS) is for training the ranking surrogate that can be used at inference to rank subsets** for each query. The  gap-index formulation helps separate sub-optimal and borderline subsets (arms) which is also the objective of gap-index frameworks like GIFA in linear stochastic bandits literature. This objective helps sample subsets that help the non-linear ranking surrogate to assign corresponding scores to such subsets with respect to rewards sampled for each val. sample and hence can tell apart good subsets from sub-optimal ones at inference time. Note here surrogate is updated in a manner to rank subsets for each validation sample as detailed in **Section 3.2**.  The gap-index top-m formulation just helps us sample  positive samples and sufficient  ambiguous samples to train this non-linear ranking surrogate in a sample efficient manner.
>
> **We request the reviewer to consider our current work GRASS for review where we have clearly stated our core reason for using gap index formulation is to train the non-linear ranking / differentiable sorting surrogate**. We believe the claims in GRASS are clear  (in submitted version) and we have also tried to clarify prior work CASE while also **clearly drawing the distinction between CASE [1] and our work GRASS**. **The reward structure, estimated mean structure and surrogate update mechanisms are completely different in GRASS compared to CASE as detailed in 3rd point here: https://openreview.net/forum?id=TsXmqOepa5&noteId=nlvTyyYJV9**.  **We DO NOT use the average validation accuracy formulation like CASE here as surrogate has to learn instance level scoring for subsets**
>
> Hence the algorithm, loss formulation for update of surrogate and the update mechanisms are **completely different** in GRASS reflecting it's use in downstream instance-level (dynamic) selection. The top-m formulation primarily helps sample enough ambiguous arms which the ranker learns to score based on ranking loss with respect to reward vector calculated for subset with respect to each validation sample.   We are happy to incorporate any suggested clarifications to make this clear. We believe the algorithmic and theoretical contributions of GRASS can also be extended to other ranking scenarios involving expensive models like LLMs where  sample-efficient is of importance.
>
> *We politely request for a fair consideration of our work with the empirical & algorithmic contributions.* We are happy to answer any further queries you may have. Our **code is also AVAILABLE on anonymous github which demonstrates our algorithm along with inference**.
>
>
> [1] Sample Efficient Demonstration Selection for In-Context Learning
> [2]  Selecting Demonstrations for Many-Shot In-Context Learning

---

> ### Author Response · Authors · 2025-11-22
> **A gentle reminder to REviewer HVfP to consider providing feedback on submitted work GRASS which employs a different algorithm than the baseline [1] for update and formulation.**
>
> Greetings. We thank you for your time and consideration. A gentle reminder to reconsider our current work. We have detailed difference between GRASS and CASE and how in GRASS the formulation for loss and surrogate update aligns with need for instance-level ranking. Note the algorithm and gap-index  formulation is also different in $GRASS_{dynamic}$. **We request the reviewer to refer to point 3 in our previous comment https://openreview.net/forum?id=TsXmqOepa5&noteId=nlvTyyYJV9 and https://openreview.net/forum?id=TsXmqOepa5&noteId=rvgzfvU8OM and we also request the reviewer to refer to Section 3.2**.
>
> **While we have tried to clarify prior work CASE [1], we politely request the reviewer to consider reviewing the algorithm and claims in current submission GRASS where the algorithm, mean estimation, LOSS and surrogate UPDATE is different from CASE.**
>
>       **To summarize CASE focuses on static selection and hence uses average accuracy as reward to update the surrogate but GRASS computes a reward vector - rewards for subsets for each validation sample, mean vector and surrogate is updated based on RANKING LOSS calculated based on ranking of subsets for each validation sample**
>
>  While we offer a static selection ablation, the surrogate update is primarily tailored for instance-level (dynamic) selection**. We  believe the evaluation assuming GRASS is entirely based on CASE is a misinterpretation and GRASS is **not mere replacement of linear surrogate with non-linear surrogate ( pointed out in Line 249) as the objectives and components of algorithm are fundamentally different**. We employ the gap-index based top-m philosophy of linear stochastic bandits (GIFA) [2]  for sample efficient training of a non-linear surrogate as it helps sample sufficient borderline arms, as exhaustive  sampling of confounding /borderline items for ranking is a challenge.
>
> *We request the reviewer to reconsider the evaluation in light of our response and request for a consideration of our current work and request feedback on the same*. We are happy to clarify further and would like to politely point out that our approach $GRASS_{dynamic}$ is **not exactly similar to philosophy to CASE or a mere replacement of linear surrogate with non-linear surrogate**.
>
>
> [1]Sample Efficient Demonstration Selection for In-Context Learning
>
>
> [2] Top-m identification for linear bandits
>
> ## **We request the reviewer to reconsider the evaluation in light of our response as we believe we have tried to clear the misunderstanding and GRASS models score of subsets at instance level  aligning with intuition of dynamic selection and not the average val. set accuracy like CASE [1]  which focuses on task-level selection. We also request for a consideration of our current work and request feedback on the same**.
>
> We are happy to clarify further and would like to politely point out that our approach $GRASS_{dynamic}$ is **not exactly similar to philosophy to CASE or a mere replacement of linear surrogate with non-linear surrogate**. We request for discussion and feedback of our current work GRASS as we believe we have outlined how GRASS differs from static selection [1] in our response and is also detailed in 3.2. Thank you for your valuable time.

---

> > ### Comment · Reviewer_HvfP · 2025-11-24
> > **Review IS UPDATED**
> >
> > Dear authors, after reading your answers, I updated and completed my review of your paper. Unfortunately, your answers did not address my toy example and did not convince me.
> >
> > Please, thoroughly analyze particular math reasoning I propose in my review for further steps.

---

> ### Author Response · Authors · 2025-11-24
> **We clarify some misunderstandings (REviewer HVfP)**
>
> Greetings,
>
> Thank you for your response. We went through the review but would again like to clarify **GRASS is fundamentally different from CASE** except for using gap-index philosophy from reda et. al. The current review still seems to reflect that GRASS does not account for dynamic selection objective and just build upon [1] which is not the case as detailed in the paper and our responses above. We politely request the reviewer to consider our clarifications.
>
> CASE objective is for static selection, our algorithm, loss and updates focus on dynamic selection. In fact, our optimization mechanism is at instance level and trains a non-linear surrogate to choose top-ranking subsets for each instance **Lines 309-313**.
>
>
> However our subset construction and optimization objective in GRASS is at instance level AND COMPLETELY DIFFERENT FROM CASE [1]. **We wanted to request if the reviewer had gone through Section 3.2 of our work which introduces the formulation and Algorithm 1 which employs different objective and formulation compared to CASE**. Hence, we believe the discussion of Equation 2 in CASE [1] is not relevant here. We also discuss the limitation of static formulation of CASE in 3.1.1 after introducing the preliminaries. Hence, we believe characterizing CASE [1] objective (equation 2) for our current work is not accurate and the optimization approach in our current work learns to rank subsets directly from LLM rewards as scores on each validation instance and not average performance like in CASE [1] (equation2). We believe this would address your queries regarding dynamic selection objective and how subsets are characterized.
>
> Also in regard to query regarding diversity, please note that the k examples in a subset are sampled from different clusters and we consider different combinations of such k examples. Hence, a subset would not just contain the closest neighbors, ensuring subsets are diverse. Also, it goes without saying, no two subsets would be exactly the same or heavily similar in the search space.
>
> Additionally,  since the search space comprises subsets with different  combinations of examples and the gap-index framework for stochastic bandits in general  samples arms with higher ambiguity (borderline arms)  and promotes uncertainty guided exploration  of diverse arms and hence does not oversample very similar arms. Note that in empirical results where the gap between ambiguous arms decreases gradually and regret also decreases gradually demonstrates that subsets are quite diverse and exhibit statistically meaningful differences (pairwise gaps).

---

> ### Author Response · Authors · 2025-11-24
> **Request for reviewer to consider the clear distinction between GRASS and CASE. Current review confuses CASE[1] objective with GRASS which is not accurate**
>
> We once again politely request the reviewer to consider the clear distinction in optimization mechanisms in GRASS and CASE. Please note that **GRASS focuses on ranking subsets per validation sample and rewards are also per-validation sample**, unlike CASE which considers  maximizing average reward on validation set. Please note that at inference time the ranker computes ranking for subsets for each test-query demonstrating our optimization in Algorithm 1 aligns with dynamic selection intuition pointed out in your response. GRASS focuses on sample efficient training of this ranker by instantiating it as a surrogate to model subset scores at val. instance level. We do not use the equation 2 and algorithm of CASE  which is tailored for static selection and we believe comparison to this objective is **not a fair representation & characterization of our current work**. We have also made this distinction in our paper and also our results discussion.
>
> We request the reviewer to acknowledge this distinction as we believe the current comparison of our work to objective in CASE is a misconception owing to the fact that GRASS uses a different ranking objective and rewards which operate at validation instance level. To further simplify the intuition of our current work, it is equivalent to sample efficient learning of a ranking function where queries are from a validation set and each subsets to be ranked represents documents. Any ranker must be trained with sufficient number of positive and negative documents per query. The gap-index formulation samples sufficient ambiguous arms (subsets) which are equivalent to hard-negatives in classical ranking setup helping our non-linear surrogate distinguish between good subsets and sub-optimal /borderline ones.

---

> ### Author Response · Authors · 2025-11-24
> **Request to REviewer HVfP to review GRASS formulation (Section 3.2) and Algorithm 1, Current review juxtaposes objective of prior static selection work with our current work though we do not use equation 2 of their work for optimization as pointed out by reviewer**
>
> Greetings,
> We apologize for the subsequent follow-ups. We believe the current review confuses CASE [1] formulation (static selection) with dynamic selection objective in GRASS. We **do not use equation 2 of [1] as objective  for dynamic selection approach of GRASS**. We believe the toy example **confuses** the effects when  CASE task-level subsets objective of equation 2 is directly extended to dynamic selection which is NOT the case as we model the problem as a learning to rank task per instance-level. Given that GRASS optimizes a ranking objective at instance (per val. query.) level we believe the limitations pointed out does not related to our proposed approach.
>
> We develop a  new algorithm based on gap-index philosophy (reda et. al [2] ) for sample-efficient training of a non-linear surrogate (ranker) which learns to rank subsets based on LLM reasoning performance for a query (instance-level). We point out the limitations of prior static selection approaches (like LENS, EXPLORA, CASE) which optimize for selection of exemplars that optimize average performance in 3.1.1 motivating need for dynamic selection leading to the formulation and algorithm in **Section 3.2**. We politely request the reviewer to evaluate our current work, given that we have made this distinction in our original work and in our responses.
>
> [2] https://arxiv.org/pdf/2103.10070

---

> ### Author Response · Authors · 2025-11-24
> **Request clarity and  to review the toy example with respect to optimization objective in our current work**
>
> Thank you again for your response. Apologies again for this follow-up. We wanted to request the reviewer to review the math construction and toy example with respect to optimization objective in our current work GRASS.  In our current work, we model subsets performance at the **level of each validation instance (not average performance)** and  the bandit optimization mechanism ensures that borderline subsets which are ambiguous are sampled sufficiently to reduce their uncertainty. These arm pulls contribute to fitting of the non-linear surrogate, learn to distinguish good arms from borderline arms and hence our optimization approach can be roughly interpreted **as a sample-efficient contrastive approach that helps the surrogate distinguish good arms and bad arms**. The surrogate learns to rank based on pairwise gap comparisons, and hence we adopt the gap-index based MAB framework. Note that each arm pull constitutes on forward pass and one backward pass of the non-linear surrogate on the dataset curated incrementally so far from arm pulls. Since, we do not have ground truth annotated subsets per instance, the gap-index based bandit algorithm helps in sample-efficient approach for fitting a non-linear surrogate through adaptive sampling of LLM rewards.
>
> Also we would like to politely point that our work focuses on dynamic setting and not hybrid setting (dynamic selection over statically selected subsets as outlined in your review. **We would like to point out that current review confuses prior work and [1] which we believe is due to misconception** that ``GRASS is just CASE with non-linear surrogate" which  **is a mischaracterization of our current work** as our rewards, surrogate updates and loss are completely different as pointed out multiple times by us in response. We request the reviewer to provide feedback and reviews reflecting the approach proposed in current work.
>
> Please note that eq2 and section 3.2. in CASE optimizes for task-level selection (not dynamic or hybrid) and hence focuses on average reward which is a similar objective & definition of average performance adopted by prior task-level selection works like  [2].   Further, we would like to **CATEGORICALLY STATE** as already mentioned in our paper and our responses that we DO NOT follow the subset scoring mechanisms in CASE [1] but **we model subset scores at instance level (per val.sample) which helps the ranker score subsets at inference time**. We are not sure how this violates intuition of dynamic selection as our bandit optimization helps merely prioritize samples based on pairwise gap comparisons to train the non-linear surrogate as exhaustive sampling of LLM rewards for all subsets w.r.t each validation sample is expensive. We also provide document ranking analogy above and how our current work can be viewed as sample-efficient learning mechanism of a ranking (diff. sort) approach.
>
> Note that optimal subsets are selected by ranking each subset with respect to test query ($GRASS_{dynamic}$) and the non-linear surrogate (ranker) is also optimized on LLM performance based rewards for each validation instance to distinguish good subsets from worse ones and **not average performance as adopted by prior work [1]**
>
> We would like to politely posit the following query: **Doesn't learning a ranker for ranking subsets at instance level directly in sample-efficient manner directly support dynamic selection intuition, as we characterize good subsets based on this instance level rewards ? Please note our loss objective and rewards are not same as CASE and our current work learns to rank good subsets higher based on instance level rewards and not average performance as adopted by equation 2 [1].**
>
> ## We would like to iterate that our current work GRASS **does not** perform optimization or fit the non-linear surrogate based on optimization loop or loss or rewards (based on average performance) of [1]   The current reviews do not unfortunately characterize the current algorithms proposed in the submitted paper but rather heavily focuses on assumption that objective of prior work is directly adopted. Hence, we are unclear how to clarify or proceed with the feedback provided. The toy example also characterizes objective of [1] and not our current work.
>
>
> [1]Sample Efficient Demonstration Selection for In-Context Learning
>
>
> [2] https://aclanthology.org/2025.findings-acl.608.pdf
>
> Kindly let us know if there are any further clarification required. We politely request for feedback in our current instance-level ranking based sample-efficient optimization approach as the current weaknesses pointed out characterize our current work as modelling the subset scores the same way as prior work [1] which **IS NOT** the case as clarified in our responses and also detailed in **Section 3.2 & algorithm 1** of our submitted version of the paper.

---

> ### Author Response · Authors · 2025-11-24
> **Request to reviewer HvfP for providing feedback reflecting objective in our current work**
>
> We would like to iterate that our current work GRASS **does not** perform optimization or fit the non-linear surrogate based on optimization loop/ loss / rewards (based on average performance) in [1]. The surrogate models the subset score based on rewards with respect to each validation instance and does not model the average reward on val. set like [1]. Respectfully, the current reviews **do not unfortunately characterize the current algorithms proposed in the submitted paper** but rather heavily focuses on assumption that objective of prior work [1]  is directly adopted which is a *major misunderstanding*. Please also refer our previous comment on how some of the strengths too *mischaracterize our approach*.  Hence, *we are unclear how to clarify or proceed with the review provided*. The toy example also focuses on arguments related to the objective of [1] and not our current work. We politely request the reviewer to refer to our prior responses for the distinction and **Section 3.2, Algorithm1** and, results ($GRASS_{dynamic}$) our result analysis for clarity.  We also explicitly state that [1] focuses on task-level selection and does not work well for dynamic selection (**Lines 59-60**, **231-234**)).
>
> Hence, we are not sure how discussion of task-level selection objective  in [1]  is relevant to dynamic selection as our optimization objective is different in our current work and is not an extension of the objective in prior work [1].  We request for feedback  reflecting the bandit based diff- sort (ranker) surrogate optimization proposed in our work which operates on LLM rewards & loss for ranking subsets at **instance level (each val. sample)** which will help us improve the quality of the work.
>
> [1]Sample Efficient Demonstration Selection for In-Context Learning

---

> ### Author Response · Authors · 2025-11-25
> **Request for clarity regarding current review to reviewer HvfP and clarification regarding new updates on diversity in the review.**
>
> We once again request clarity regarding discussion of the objective with respect to our current work as we do not use the optimization. objective in [1]. We politely request for review of our current optimization algorithm that operates with rewards sampled at instance level (319-320) discussed in our work.
>
> We are also unclear by the new comment in the review :
>
> ``UPD: In their responses, authors substitute the diversity of
> subsets ("arms") in the set, what was my main question, by the diversity of exemplars in one subset".
>
> We would like to point out that a **part of our previous response directly addresses the diversity of m subsets**. From what we understand, the reviewer mentions top-m subsets could be near duplicates of each other in sense arms could be heavily correlated or redundant ( near duplicates of each other).
>
> We would like to again point out that the gap-index algorithm by design ensure exploration of diverse arms and pairwise gap comparisons help debug the scenario presented by the reviewer. Note if arms (subsets) in search space are near duplicates the confidence radii of arms overlaps heavily causing collapse in gaps, but our empirical results demonstrate **gradual decrease in gaps across rounds** (**Figures  2a 3a**) and regret demonstrating this is not the case as outlined in our previous response too where we point to the empirical evidence and gap based bandit algorithm's intuition that helps debug scenarios posited by the reviewer. This shows that different subsets in space are not near duplicates and exhibit meaningful (pairwise gap-based) differences.
>
> Also whether two arms are heavily similar to each other depends on the constituting examples. We ensure the subsets are not near redundant or exactly same in terms of constituting examples and also consider different  combinations of examples.
>
> We also refer the reviewer to ranked subsets, selected dynamically at query level by our ranker at this link in repo (https://anonymous.4open.science/r/top-m-arm-selection-non-linear-C010/data/AQUA_RAT/GRASS_aquarat_lalam_3_2_test_examplars.json) we have linked in our paper to demonstrate that ranked subsets  per query are *not redundant* (note that in the file first key "1" denotes first test sample & so on each with a dict representing different components (question, answer, optionally rationale) of the examples in subset).
>
> We also refer the reviewer to format of some of  training samples in our data folder (https://anonymous.4open.science/r/top-m-arm-selection-non-linear-C010/data/AQUA_RAT/train_aqua_mistral3.txt) which was adaptively collected by our bandit algorithm to train the ranking surrogate which would demonstrate our optimization process (ranking performed at instance - i.e. each query level).
>
> We hope we have clarified with examples and algorithmic intuition of how subsets ranked are not redundant during training or inference. We also **politely request the reviewer to provide reviews and feedback on  our current work GRASS** as we have tried our best to make the distinction between prior work [1] and our work. **We would like to iterate that our current work GRASS does not perform optimization or fit the non-linear surrogate based on optimization loop or loss or rewards (based on average performance) of [1]**, but current review focuses on this argument regarding optimization objective in [1] **which is not** related to and misrepresents our current work which focuses on ranking subsets for instance-level ( per validation sample/query).
>
> >   **We respectfully and politely convey that the current review & evaluation unfortunately does not represent the optimization & approach in our current work** as mentioned in previous comments and pointers to specific sections in the paper where we already make the distinction. The current review and evaluation *does not align with / reflect our current work* and we are unclear on next steps or feedback to act upon.
>
>  We *look forward to the discussion* of the optimization of ranking surrogate **per instance (validation query) level**  adopted in the **submitted work**, aligning with it's usage to rank subsets per test query at inference time and are happy to clarify any queries in this regard.
>
> With less than a week remaining for discussion period, we politely request the reviewer to consider this request regarding alignment of review to current work submitted  as it would help in productive discussion and valuable feedback for progression of the submitted work for instance-level (dynamic) selection.
>
> [1]Sample Efficient Demonstration Selection for In-Context Learning

---

> ### Author Response · Authors · 2025-11-26
> **Request to reviewer HvfP**
>
> We request the reviewer to evaluate the conceptual limitation pointed out in review as we have in our response and in our current work it is explicit that our work GRASS **does not follow equation 2 of prior work [1] or neither exactly the formulation as [1] focuses on task-level selection** and our work instead employs  instance level ( val query)  rewards and focuses on **dynamic selection** (please refer our paper and comments above) for learning to rank subsets as detailed in previous responses.
>
> We have also answered the **diversity  query in our previous comment** & earlier response too with pointers to empirical analysis and algorithmic intuition. We do not substitute the notion of diversity as mentioned in updated review and we request the reviewer to read our previous response in detail on how the diversity of top-m subsets can is verified. We also added pointers to qualitative examples in repo showing diversity of top-m subsets.
>
> We **politely request the reviewer to review our current work** as the current conceptual limitation and other aspects discussed in the review **does not unfortunately align with or discuss optimization loop & objective of current  work submitted to this conference**.   We also explicitly state that [1] focuses on task-level selection and does not work well for dynamic selection (Lines 59-60, 231-234)) and hence we are not sure why its being **assumed we are using eq2 of [1] when we have explicitly stated how task level selection is limited and also all our reward calculations and loss (**lines    (319-320), eq 3** operates at instance level and is fundamentally different from [1]**. Discussing the current work would also help us improve quality of our work and we are happy to provide any further clarifications. We request the reviewer to provide a revised review aligning with current work. We also politely request for reassessment of evaluation and scores in light of this information.
>
> [1]Sample Efficient Demonstration Selection for In-Context Learning

---

> ### Author Response · Authors · 2025-11-27
> **Request to reviewer HvfP for review of current work and review of clarification reg. diversity of subsets**
>
> We request the reviewer to evaluate the conceptual limitation pointed out in review as we have in our response and in our current work it is explicit that our work GRASS **does not follow equation 2 of prior work [1] or neither exactly the formulation as [1] focuses on task-level selection** and our work instead employs  instance level ( val query)  rewards and focuses on **dynamic selection** (please refer our paper and comments above) for learning to rank subsets as detailed in previous responses.
>
> We have also answered the diversity  query in our previous comment & earlier response too with pointers to empirical analysis and algorithmic intuition. We also added pointers to qualitative examples in repo showing diversity of top-m subsets.
>
> We **politely request the reviewer to review our current work** as the current conceptual limitation and other aspects discussed in the review **does not unfortunately align with or discuss optimization loop & objective of current  work submitted to this conference**.   We also explicitly state that [1] focuses on task-level selection and does not work well for dynamic selection (Lines 59-60, 231-234)) and hence we are not sure why its being **assumed we are using eq2 of [1] when we have explicitly stated how task level selection is limited and also all our reward calculations and loss (**lines    (319-320), eq 3** operates at instance level and is fundamentally different from [1]**. Our dynamic approach also **does not** just operate over the **task-level** selected sets during inference like  static/hybrid variants in prior work [1].  Our proposed approach $GRASS_{dynamic}$ is not in anyway directly based on or optimizing for average accuracy like in **eq 2** of prior work [1] and we request the reviewer to review the current response,  our prior responses and Section 3.2 (pointers: (**lines    (319-320), eq 3**) of **current work** submitted to this conference to confirm the same.
>
> We are happy to clarify any queries in this regard and request a discussion, as we believe the current limitation raised is due to a **major misconception of our work**. We have stated the major difference and how we are not optimizing for **eq 2** and also our features, rewards estimation and loss fundamentally differ from prior work [1]. We are happy to clarify specific queries the reviewer may have on any component of current work.
>
> Discussing the current work would also help us improve the quality of our work and we are happy to provide any further clarifications. We request the reviewer to provide a revised review aligning with current work. We believe the **current conceptual limitation** posed in review is for an approach that **does NOT exist in our submitted work** as we explicitly state limitations of task-level selection approach and adopt a dynamic selection approach that learn a ranking model in sample-efficient manner. We also politely request for reassessment of evaluation and scores in light of this information.
>
> [1]Sample Efficient Demonstration Selection for In-Context Learning

---

### Official Review · Reviewer_XXN1 · 2025-11-01

**Soundness:** 2
**Presentation:** 1
**Contribution:** 2
**Rating:** 2
**Confidence:** 2

**Summary:**

The paper introduces GRASS, a novel framework that addresses the top-m arm selection problem, e.g., selecting optimal demonstration subsets for LLM in-context learning. The primary contribution is the integration of a non-linear ranking surrogate based on differentiable sorting within the existing gap-index bandit framework, which explicitly models the complex, non-linear reward, a limitation of prior linear-based approaches. GRASS provides an efficient solution for both static (task-level) and dynamic (instance-level) subset selection, yielding significant performance gains and faster convergence compared to existing state-of-the-art methods.

**Strengths:**

- The paper introduces a novel non-linear approach for exemplar selection in in-context learning, supported by theoretical analysis on the gap index.
- GRASS significantly improves the performance of in-context learning while yielding comparable efficiency.

**Weaknesses:**

- The writing could be improved; it's very difficult for readers who are not familiar with multi-arm bandit for LLM in-context learning to understand the approach. I'd suggest putting more details in the problem definition, including step-by-step examples of sample selection with multi-arm bandit, notations, and important definitions such as demonstrations, empirical mean estimator, gap index, rewards, etc.
- If I understand correctly, the main contribution and also the difference between GRASS and CASE is the non-linearity, which is the differentiable sorting model with MLP in Eqn. 2. The novelty seems limited, and the motivation for non-linearity is unclear to me.
- Section 3.3 should have a Remark paragraph explaining the theoretical results and their implication.
- The paper only conducts experiments on two models, llama 3b and gpt 4o. Evaluating more models will strengthen the claim and effectiveness of GRASS.
- Minor: missing argument in the definition of gap-index (end of Page 5) and format problem at the end of Page 6.

**Questions:**

Please see Weaknesses.

---

> ### Author Response · Authors · 2025-11-17
> **Response to  Reviewer XXN1 (1/2)**
>
> We thank the reviewer for their valuable feedback. Below we address some of the queries.
>
> Q1. *The writing is difficult for readers who are not familiar with multi-arm bandit for LLM ICL. I'd suggest putting more details in the problem definition, including step-by-step examples of sample selection with multi-arm bandit, notations, and important definitions such as demonstrations, empirical mean estimator, gap index, rewards, etc.*
>
> **Response**: Thank you for this feedback to improve the quality of our work. We would like to politely point out that we have already defined demonstrations / examples in **Lines 42 and Lines 171-172** in the submitted version of the paper, rewards have been defined in  **Lines 239-240** and more formally defined in Equation 3 and **Lines 300_301**, gap-index in Line 311.
>
> **Changes**: We have now attached a Figure (**Figure 1**) in the *revised version of the paper* which provides an overall idea of our example subset selection algorithm GRASS and how $GRASS_{dynamic}$ inference is performed. We hope this clarifies the MAB approach for improving ICL performance of LLMs. Further, Lines **302-323** give a step-by-step overview of Algorithm 1 including details of how the arm is pulled, reward is sampled, and the non-linear surrogate is updated using the ranking loss at **instance level**.
>
> Q2. *If I understand correctly, the main contribution and also the difference between GRASS and CASE is the non-linearity, which is the differentiable sorting model with MLP in Eqn. 2. The novelty seems limited, and the motivation for non-linearity is unclear to me.*
>
> **Response**:  We appreciate the reviewer’s question regarding the novelty of the non-linear scoring model. We would like to clarify that the contribution is non-trivial and not just replacement of linear surrogate with a non-linear model, rather, the *non-linear reward model*, dynamic setup ,*fundamentally changes the structure of the problem** and requires a *new theoretical and algorithmic treatment* compared to CASE.
>
> Introducing non-linearity breaks key assumptions underlying theoretical framework of CASE:
> -  CASE relies on linear estimators with closed-form confidence bounds. These assumptions break in our setting
> -  CASE cannot propagate uncertainty through a non-linear differentiable sorting pipeline;
> -  CASE gap estimation and corresponding theoretical bounds does not extend to non-linear surrogates, dynamic setup and hence requires algorithmic modifications including new gap estimation, loss, surrogate update mechanism along with new theoretical bounds on pairwise gaps.
>
> In GRASS, we contribute:
> - **New gap-estimation, loss, surrogate update mechanism & theoretical results**:  Changes in mean estimation and gap-index estimation as described in - **Algorithm 1** and **Section 3.2**.  Definition 1, Theorem 1 (Section 3.3) for bound on pairwise gap errors.
>
> - **A new good-event construction** : enabling control of pairwise non-linear score gaps;
> - **A new sample complexity bound**: -  Theorem 2 for adaptive selection under non-linear surrogate and  uncertainty;
> - **Algorithmic modifications**: to the challenger and best arm candidate comparison strategy as outlined in Algorithm 1 and Section 3.2 because linear confidence sets are no longer valid and also requires modeling of subset scores with respect to each validation instance to align with dynamic selection.
>
> These developments are not present in CASE and cannot be derived by simply *adding an MLP*. It requires change in fundamental calculations of gap-indices, algorithm (loss, mean estimation and surrogate update mechanisms) and also theoretical results.
> In summary, the **novelty of GRASS** is not the differentiable sorting model itself, but:
>
> -  integrating non-linear reward learning with a gap-index based algorithm to fit a non-linear surrogate that sufficiently learns to distinguish good arms (subsets) from borderline ones.
> - providing the sample-complexity guarantee for non-linear surrogate based subset scoring under MC-dropout uncertainty
> -  and extending the gap-index analysis of CASE to a non-linear, data-dependent scoring environment.
>
> This amounts to a non-trivial extension both **theoretically and algorithmically**
>
> Q3. *Section 3.3 should have a Remark paragraph explaining the theoretical results*
>
> **Response**:  We would like to politely  point out that we have already added a summary at end of **Section 3.3**  in the original submission,  describing how theorem 2 and Lemma 1 together provide a high probability upper bound on sample complexity. We have also described this translates to approximately expected  number of LLM calls needed for top-m subsets identification. We have *further highlighted this and expanded the intuition* behind the theoretical results in the revised version of the paper **Lines 361-369**.
>
> Q4. *Minor issues and typo*
>
> **Response**: Thank you. We have fixed these minor issues in revised version of the paper.

---

> ### Author Response · Authors · 2025-11-23
> **Response to reviewer XXN1 (2/2)**
>
> Q5. *The paper only conducts experiments on two models, llama 3b and gpt 4o. Evaluating more models will strengthen the claim and effectiveness of GRASS.*
>
>
> **Response**:
>
> **Regarding LLMs**:We evaluate on an open model (LLama 3.2, 3B) and a larger  closed model gpt-4o-mini which are currently stable models used by the community to show our approach works for model of different parameter scales. We would like to point out that evaluation employing  multiple LLMs in current landscape is difficult due to compute requirements and costs. We also evaluate on multiple benchmarks across different domains to show that our approach works across models for diverse tasks.   We also open source our bandit algorithm to enable research using future models.
>
> *Based on your feedback*, we also **included DeepSeek-r1:7b** in our evaluation and the results are attached below. We also included them in revised version of the paper (**Table 3**).
> ## Results across datasets using Deepseek-R1:7B (5-shot for all methods)
>
> | **Method**           | **GSM8K** | **AquaRat** | **WMT** |
> |----------------------|-----------|-------------|---------|
> | **Task level**       |           |             |         |
> | EXPLORA (Purohit et al., 2024) | 82.63 | 68.10 | 78.59 |
> | LENS (Li et al., 2023)          | 77.33 | 57.87 | 76.98 |
> | Static CASE                      | 83.09 | 69.29 | 78.26 |
> | $GRASS_{static}$               | 86.12 | 70.47 | 79.27 |
> | **Instance level**   |           |             |         |
> | CASE\(_{dynamic}\) (CASE)        | 85.98 | 69.68 |  79.78   |
> | **$GRASS_{dynamic}$**  | **90.20** † | **74.41** † |   **81.50**  |
>
> **Caption:** Results across datasets using Deepseek-R1:7b (5-shot for all methods).
> † indicates statistical significance (t-test) over CASE\(_{dynamic}\) at the 0.05 level.
>
> We observe that while exemplar selection does lead to gains even with models like DeepSeek, on closer examination we observe that Deepseek follow the style and function of reasoning traces from they are fine-tuned on than those provided in in-context learning.  We particularly find the reasoning traces to be long owing to GRPO based training of these models to follow a certain style for reasoning.
>
> Overall we demonstrate sample efficient training of a non-linear surrogate for  dynamic selection can help improve performance for ICL in diverse LLMs. Our code is also released and can be easily applied to newer LLMs in future.
>
> We hope we have sufficiently addressed your queries. Kindly let us know if you have any further concerns. We look forward to addressing them. We also *politely request for reconsideration of evaluation of our work in light of the response*. Thank you for your valuable time and feedback.

---

> > ### Author Response · Authors · 2025-11-25
> > **A gentle reminder for consideration of our response and revision to Reviewer XXN1**
> >
> > Greetings Reviewer XXN1,
> >
> >  Thank you for your valuable time and feedback. Please find our responses above. In summary, we have incorporated the suggested changes to improve the presentation and additional experiments. We have added Figure 1 to explain the training of non-linear ranking surrogate using MAB for ICL and also the dynamic selection using this surrogate using inference.  We have uploaded the revised version too. We would like to request if we have addressed your concerns. Kindly let us know if you have any queries. **Thank you for your valuable time and feedback**. *We request for reassessment of scores in light of our response and revisions.*

---

### Official Review · Reviewer_kXWB · 2025-11-01

**Soundness:** 3
**Presentation:** 3
**Contribution:** 3
**Rating:** 6
**Confidence:** 3

**Summary:**

This paper addresses **top-$m$ subset selection for in-context learning (ICL)** with LLMs, formulated as a fixed-confidence multi-armed bandit problem where each $k$-sized demonstration subset is an arm. The challenge is identifying the best $m$ subsets from an exponentially large space while minimizing expensive LLM queries.

The authors propose **GRASS (Gap-indexed bandits with RAnking-based non-linear Surrogate for Selection)**, which extends gap-index bandits to use a **differentiable sorting neural network** instead of linear surrogates. Prior work (CASE) assumes linear relationships between features and rewards, failing to capture complex interactions between examples in a subset. GRASS trains the neural surrogate online using ranking loss ($-$NDCG), updating it incrementally (one epoch per round) as the bandit algorithm tests arms.

**Key technical contribution**: Adapting gap-index confidence bounds for non-linear surrogates, accounting for MC-dropout variance, SGD stability, and model bias (Theorem 1), with sample complexity guarantees (Theorem 2). The algorithm maintains a top-$m$ set and challenger list, iteratively testing the most ambiguous pairs and retraining the surrogate.

**Strengths:**

**Originality:** The paper makes a genuine technical contribution by extending gap-index bandits to non-linear surrogates with valid theoretical guarantees. Prior work (CASE) is limited to linear models, which is genuinely restrictive for modeling LLM rewards. The key insight—using differentiable sorting networks within a bandit framework and deriving modified confidence bounds that account for neural network uncertainty (MC-dropout variance, SGD stability, model bias)—is non-trivial.

**Quality:** The experimental design is solid with evaluation across three diverse datasets (math reasoning, translation) and multiple model sizes. The comparison to relevant baselines is comprehensive, and the inclusion of both static and dynamic settings demonstrates practical versatility.

**Clarity:** The paper is well-structured with clear motivation and intuitive explanations. Algorithm 1 is easy to follow, and the learning-to-rank framing is well-articulated. The problem formulation (Section 3.1) clearly distinguishes static vs. dynamic selection and positions the work relative to prior bandit formulations.

**Significance:** The problem is practically relevant—ICL example selection is important for LLM deployment, and non-linear modeling is clearly more appropriate than linear assumptions for LLM rewards. The 9-15% improvements on smaller models and 2× speedup over CASE demonstrate real value. The framework's generality (applicable beyond ICL to any subset selection problem) broadens impact. Providing both offline top-$m$ identification and a trained ranking model is useful for practitioners.

**Weaknesses:**

**Limited Experimental Scope and Missing Ablations:** The experiments are restricted to one primary open-source LLM (Llama3.2-3B) with GPT-4o-mini results relegated to the appendix. This is insufficient to validate the claim that non-linear surrogates are universally better—perhaps the gains are model-specific. It would be interesting to see results on reasoning models.

**Scalability Concerns Not Addressed:** The paper claims dynamic selection has "negligible latency overhead" (Figure 1d, ~few milliseconds), but this analysis is incomplete. Figure 1d only shows average inference time, not how it scales with problem size. For dynamic selection, GRASS must score **all** $\binom{|Q|}{k}$ possible subsets per test query. With $|Q|=100$ and $k=5$, that's ~75 million subsets. Even if the neural network is fast (say, 1ms per forward pass with batching), this becomes prohibitive. The paper never discusses: (1) How many subsets were actually ranked in experiments? (2) What happens when $|Q|$ grows to 1000 or 10,000? (3) Can approximate nearest-neighbor search or pruning strategies reduce this cost? The claim of "efficient instance-level selection" is overstated without addressing exponential growth.

**Insufficient Comparison to State-of-the-Art ICL Methods:** The baselines are primarily other bandit methods, but the paper ignores recent specialized ICL example selection methods. For instance, retrieval-augmented approaches using dense embeddings  are only briefly mentioned. The comparison to KNN and MMR is superficial—these are implemented as simple baselines without tuning. More sophisticated methods like learned retrievers are missing entirely. The claim that GRASS achieves "state-of-the-art" is therefore not substantiated.

**Questions:**

see weakness above

---

> ### Author Response · Authors · 2025-11-20
> **Response to Reviewer kXWB (1/2)**
>
> Thank you for your valuable feedback and time. We would like to address some of the queries below.
>
> Q1. Scalability concerns: The paper claims dynamic selection has "negligible latency overhead" (Figure 1d, ~few milliseconds), but this analysis is incomplete. Figure 1d only shows average inference time, not how it scales with problem size. For dynamic selection, GRASS must score all
>  possible subsets per test query. With
>  and
> , that's ~75 million subsets. Even if the neural network is fast (say, 1ms per forward pass with batching), this becomes prohibitive. The paper never discusses: (1) How many subsets were actually ranked in experiments? (2) What happens when
>  grows to 1000 or 10,000? (3) Can approximate nearest-neighbor search or pruning strategies reduce this cost? The claim of "efficient instance-level selection" is overstated without addressing exponential growth
>
> **Response**: We would like to clarify that with regard to efficiency, our primary claims are in **sample efficiency** which measures the number of LLM calls (bandit rounds) required for our proposed approach compared to existing bandit based selection methods like CASE. This is illustrated in **Figure 2c** where GRASS converges faster and also explained in **Lines 506-516**.
>   In regard to latency, We primarily show in **Figure 2d** that the non-linear surrogate ($GRASS_{dynamic}$) does not add much latency relative to inference using top-m subsets from CASE. In $GRASS_{dynamic}$ the subsets considered for ranking is the union of $U_{1..T}$ (top-m arms from rounds of GRASS) and all challenger shortlists $[C_1…C_T]$, across the bandit rounds of GRASS. Hence, even when sample size grows, it would not lead to exponential blowup at test time, as it is handled by our algorithm GRASS which performs principled sampling during subset selection stage which performs top- identification simultaneously with/update of the non-linear surrogate. Hence, we only use the candidate shortlists ($U_{1..T}$ and $[C_1…C_T]$) for ranking at inference time, where $T$ is the last time-step before convergence of the algorithm. Hence, even with more examples, the candidate subsets considered are manageable. We have clarified the subsets considered for ranking at test time in Section 4 in the revised manuscript.
>
> Q2. Insufficient Comparison to State-of-the-Art ICL Methods: The comparison to KNN and MMR is superficial—these are implemented as simple baselines without tuning. More sophisticated methods like learned retrievers are missing.
>
> **Response**:  We would like to clarify that for MMR we tuned the $\lambda$ parameter on validation set. We also used the best embeddings models for KNN and MMR as recommended in the respective research [1,2]. Regarding *learned retrievers* we would like to politely point out that in instance level selection baselines (**Table 1**), we evaluate well known  learning to rank approaches where ranking models are **trained following best practices**, using principled and diverse ranking losses as shown in **Table 4**. These baselines qualify as ** learned rankers** for enhancing in-context learning in LLMs. Additionally, since our approach frames exemplar selection as a learning to rank top-m subsets problem and includes efficient optimization for the same, we compare with related state-of-the-art bandit based selection approaches and *learned ranking models* for a fair comparison.
>
> [1] Ohad Rubin, Jonathan Herzig, and Jonathan Berant. Learning to retrieve prompts for in-context learning. In Proceedings of the 2022 Conference of the North American Chapter of the Association for Computational Linguistics: Human Language Technologies, pp. 2655–2671, Seattle, United States, July 2022. Association for Computational Linguistics. doi: 10.18653/v1/2022.naacl-main. 191
>
> [2]  Xi Ye, Srinivasan Iyer, Asli Celikyilmaz, Veselin Stoyanov, Greg Durrett, and Ramakanth Pasunuru. Complementary explanations for effective in-context learning. In Findings of the Association for Computational Linguistics: ACL 2023, pp. 4469–4484, Toronto, Canada, July 2023b. Association for Computational Linguistics.
>
> Regarding additional LLMs we are adding experiments with another LLM and we will be uploading the results shortly by today. Meanwhile, we hope we have addressed the other queries, and we request for any other queries you may have.

---

> ### Author Response · Authors · 2025-11-23
> **Response to Reviewer kxWB (2/2)**
>
> Q2. Limited Experimental Scope and Missing Ablations: The experiments are restricted to one primary open-source LLM (Llama3.2-3B) with GPT-4o-mini results relegated to the appendix. This is insufficient to validate the claim that non-linear surrogates are universally better—perhaps the gains are model-specific. It would be interesting to see results on reasoning models.
>
> **Response**: Thank you for your feedback. We would like to politely point out that we have performed ablations for proposed GRASS method as shown in Table 1. We have a version of ($GRASS_{dynamic}$) without exploration demonstrating the need for our proposed gap-index algorithm so the non-linear ranking diff.sort surrogate  learns from sufficient borderline subsets. We also have enhanced version of CASE ($CASE_{dynamic}$) as a baseline where we apply the bandit algorithm online for fair comparison to our approach.
>
> **Regarding LLMs**:We evaluate on an open model (LLama 3.2, 3B) and a larger  closed model gpt-4o-mini which are currently stable models used by the community to show our approach works for model of different parameter scales. We would like to point out that evaluation employing  multiple LLMs in current landscape is difficult due to compute requirements and costs. We also evaluate on multiple benchmarks across different domains to show that our approach works across models for diverse tasks.   We also open source our bandit algorithm to enable research using future models.
>
> *Based on your feedback*, we also **included DeepSeek-r1:7b** in our evaluation and the results are attached below. We also included them in revised version of the paper (**Table 3**).
> ## Results across datasets using Deepseek-R1:7B (5-shot for all methods)
>
> | **Method**           | **GSM8K** | **AquaRat** | **WMT** |
> |----------------------|-----------|-------------|---------|
> | **Task level**       |           |             |         |
> | EXPLORA (Purohit et al., 2024) | 82.63 | 68.10 | 78.59 |
> | LENS (Li et al., 2023)          | 77.33 | 57.87 | 76.98 |
> | Static CASE                      | 83.09 | 69.29 | 78.26 |
> | $GRASS_{static}$               | 86.12 | 70.47 | 79.27 |
> | **Instance level**   |           |             |         |
> | CASE\(_{dynamic}\)       | 85.98 | 69.68 |  79.78   |
> | **$GRASS_{dynamic}$**  | **90.20** † | **74.41** † |   **81.50**  |
>
> **Caption:** Results across datasets using Deepseek-R1:7b (5-shot for all methods).
> † indicates statistical significance (t-test) over CASE\(_{dynamic}\) at the 0.05 level.
>
> We observe that while exemplar selection does lead to gains even with models like DeepSeek, on closer examination we observe that Deepseek follow the style and function of reasoning traces from they are fine-tuned on than those provided in in-context learning.  We particularly find the reasoning traces to be long owing to GRPO based training of these models to follow a certain style for reasoning.
>
> Overall we demonstrate sample efficient training of a non-linear surrogate for  dynamic selection can help improve performance for ICL in diverse LLMs. Our code is also released and can be easily applied to newer LLMs in future.
>
> We hope we have sufficiently addressed your queries. Kindly let us know if you have any further concerns. We look forward to addressing them. We also *politely request for reconsideration of evaluation of our work in light of the response*. Thank you for your valuable time and feedback.

---

### Author Response · Authors · 2025-11-23
**General Response to Reviewers and Summary of Responses**

We would like to thank the reviewers and the area chair for their valuable time. We have submitted individual responses to queries from reviewers and also uploaded a revised version. We would like to summarize the concerns raised and our responses:

1. Response to **Reviewer kXWB**: With regards to queries regarding ablations and experiment on additional LLMs - we clarified the ablations already present in the submitted version of the paper. Additionally, apart from the two LLMs we already experimented on, we added results for an open reasoning-based distilled  model Deepseek-R1:7B and included the results in our response and revised version of the paper **Table 3**.  We also clarified how the learning to rank baselines account for learned retrievers mentioned by the reviewer. We also clarified and responded to scalability concerns for ranking at inference time and regarding sample-complexity for the bandit algorithm run to fit the non-linear surrogate.

2. Response to **Reviewer XXN1**: The query regarding additional LLM experiments were addressed as mentioned above (Table 3 in revised paper). We also added a Figure (Fig.1 ) to better convey the flow of our approach. Additionally, we also clarified the misunderstanding that GRASS is just an extension of CASE. **GRASS is not merely an extension of CASE**, replacing the linear surrogate with a non-linear MLP. It requires fundamental changes in the algorithm such as **loss objective, surrogate update mechanisms, mean estimation, gap estimation**, **new gap-index analysis, new pairwise gap bounds and sample-complexity bounds** to account for fitting a non-linear surrogate for dynamic selection.


3. Response to **Reviewer HvfP**: The conceptual limitation  mentioned by Reviewer HvfP is a misrepresentation & review is **NOT representative of or aligned to the submitted work**. For instance, we do not use the formulation (eq 2) of CASE[1], mentioned by reviewer in submitted work.

 The comments in review are primarily due to **major misconceptions** as follows:

- **GRASS is just building upon CASE replacing the linear surrogate with non-linear one**:  We would like to clarify that **GRASS is not an extension of CASE with non-linear surrogate**. **GRASS** introduces a fundamentally different algorithm including **changes to loss objective, rewards, surrogate update** mechanism to account for fitting a non-linear surrogate that can be employed as ranker for dynamic selection during inference time. These details are presented in **Section 3.2** (particularly **Lines 309-314**) and limitations of static (task-level) subsets from approaches like [1] are discussed in 3.1.1.  Hence, we are not sure how discussion of task-level selection objective in [1] is relevant as our optimization objective is different & not an extension of objective in [1]. The details are also in our response to Reviewer HvfP.

- **Prior work (baseline) CASE [1]  formulation is intended for dynamic selection**: The prior work CASE [1] is **intended for static (task-level) and not dynamic selection** and the goal to select top-m subsets with higher task-level performance estimates than sub-optimal ones matches intuition of prior works focusing on task-level selection [2]. Hence, the objective of [1] to find subsets with higher performance than other subsets to prune sub-optimal sets from consideration. Also **we do not use the eq 2 objective of [1] and we are not sure why the review focuses discussion on the same** as our goal is dynamic selection. Our objective and algorithm is different as briefly described above and in our response.

To summarize **CASE focuses on static selection** & hence uses average accuracy to update the surrogate but GRASS uses a **instance-level RANKING LOSS** as detailed in **Lines 309-313** is w.r.t mean & reward vectors for each val. sample.  Hence, surrogate is updated w.r.t scores assigned to subsets with respect to each validation sample *aligning with needs of dynamic subset selection*.

*We request reviewer HvfP & Area Chair to consider this distinction*, which would demonstrate that the dynamic selection intuition is **NOT violated** as the non-linear surrogate learns to rank subsets at instance (each val. sample) level. We use the gap-index formulation as it sufficiently samples ambiguous arms (subsets) which helps fit the non-linear surrogate and equip it with capability to separate confounding (borderline) subsets from good subsets. We request the reviewer to re-evaluate in light of this clarification and we also **politely request for the reviewer's feedback on our current submitted work**. The current review **DOES NOT align** with the optimization objective of our **current work** as *clarified in our response pointing to details in the paper.*

[1] Sample Efficient Demonstration Selection for In-Context Learning
[2] Selecting Demonstrations for Many-Shot In-Context Learning

We hope we have addressed all the concerns of reviewers adequately. Thank you.

---

### Meta-Review · Area_Chair_dPkX · 2026-01-07

**Summary:**

The paper proposed a method, GRASS, that selects optimal subsets of demonstration for LLM In Context Learning (ICL). GRASS extends the linear surrogate function with a non-linear MLP and proposes a learning algorithm to account for such non-linear surrogate. GRASS demonstrates significant performance for both static (task-level) and dynamic (instance-level) subset selection in experiments.

The reviewers are primarily concerned about the lack of rigorous empirical evaluations (diverse set of LLMs, comparisons to SOTA baselines) and contributions compared to the previous work, as well as explanations on various sections of the paper. Upon reading both the reviews/discussions and the paper, I also think that the experimental evaluation in the paper is still limited, especially the paper's motivation for non-linear surrogate is based on LLMs, and the paper’s analysis/clarity could be significantly improved.

**Reviewer Concerns:**

Reviewer kXWB is concerned about the limited experiment (2 LLMs, and computational overhead, SOTA baselines for ICL).
Reviewer XXN1 is concerned about the writing, which is difficult for the general readers, and limited novelty, as well as limited experiments (2 LLMs).
Reviewer HvfP is concerned about the problem formulation and differences from Purohit et al's work. The reviewer also raised potentially incorrect claims in the paper.

The authors have clarified most concerns about the clarity of the paper, including making the paper clearer, and provided new experimental results on LLMs. Nevertheless, I also think that the paper's presentation can still be significantly improved, and the experimental evaluation is still lacking.

**Reviewer Scores:**

- kXWB: 6 - Did not respond
- XXN1: 2 - Did not respond
- HvfP: 2 - Responded but still concerned about the clarity on contributions and claims of the paper.

---

### Decision · Program_Chairs · 2026-01-26

Reject